*Report*

# Immunodynamics of explanted human tumors for immuno-oncology

Agathe Dubuisson[1,2,†] [iD], Jean-Eudes Fahrner[1,2,3,†] [iD], Anne-Gaëlle Goubet[1,2,†], Safae Terrisse[1,2],
Nicolas Voisin[1,2], Charles Bayard[1,2], Sebastien Lofek[1,2], Damien Drubay[1,4,5], Delphine Bredel[1,2],
Séverine Mouraud[1,2] [iD], Sandrine Susini[1,2], Alexandria Cogdill[1,2,6], Lucas Rebuffet[1,2], Elise Ballot[7,8],
Nicolas Jacquelot[1,9,10], Vincent Thomas de Montpreville[11] [iD], Odile Casiraghi[12], Camélia Radulescu[13],
Sophie Ferlicot[14], David J Figueroa[15], Sapna Yadavilli[15], Jeremy D Waight[15], Marc Ballas[15], Axel Hoos[15],
Thomas Condamine[16], Bastien Parier[17], Christophe Gaudillat[18], Bertrand Routy[19,20],
François Ghiringhelli[7,8,21], Lisa Derosa[1,22], Ingrid Breuskin[23], Mathieu Rouanne[1,12,24,25],
Fabrice André[1,26], Cédric Lebacle[17], Hervé Baumert[1], Marie Wislez[27,28], Elie Fadel[29], Isabelle Cremer[30,31],
Laurence Albiges[1,22], Birgit Geoerger[32] [iD], Jean-Yves Scoazec[1,12], Yohann Loriot[1,22],
Guido Kroemer[1,33,34,35,36,37] [iD], Aurélien Marabelle[1,38], Mélodie Bonvalet[1,39] & Laurence Zitvogel[1,2,26,39,*] [iD]

1    Institut Gustave Roussy, Villejuif, France
2    Institut National de la Santé Et de la Recherche Médicale (INSERM) U1015, Equipe Labellisée—Ligue Nationale contre le Cancer, Villejuif, France
3    Transgene S.A, Illkirch-Graffenstaden, France
4    Service de Biostatistique et d'epidémiologie, Gustave Roussy, Université Paris-Saclay, Villejuif, France
5    Oncostat U1018, Inserm, Université Paris-Saclay, Équipe Labellisée Ligue Contre le Cancer, Villejuif, France
6    The University of Texas MD Anderson Cancer Center, Houston, TX, USA
7    Cancer Biology Transfer Platform, Centre Georges-François Leclerc, Dijon, France
8    Centre de Recherche INSERM LNC-UMR1231, Dijon, France
9    Walter and Eliza Hall Institute of Medical Research, Melbourne, VIC, Australia
10   Department of Medical Biology, University of Melbourne, Melbourne, VIC, Australia
11   Service d'Anatomie et cytologie pathologiques, Hôpital Marie Lannelongue, Le Plessis-Robinson, France
12   Departement de Biologie et Pathologie Médicales, Gustave Roussy Cancer Campus, Villejuif, France
13   Service d'Anatomie et cytologie pathologiques, Hôpital Foch, Suresnes, France
14   Service d'Anatomie et cytologie pathologiques, Hôpital Bicêtre, Le Kremlin-Bicêtre, France
15   Oncology R&D, GlaxoSmithKline, Collegeville, PA, USA
16   Incyte Research Institute, Wilmington, DE, USA
17   Service de Chirurgie urologique, Hôpital Bicêtre, Le Kremlin-Bicêtre, France
18   Service d'Urologie, Hôpital Saint Joseph, Paris, France
19   Division of Oncology, Department of Medicine, Centre de recherche du Centre hospitalier de l'Université de Montréal (CRCHUM), Montréal, QC, Canada
20   Hematology-Oncology Division, Department of Medicine, Centre Hospitalier de l'Université de Montréal (CHUM), Montréal, QC, Canada
21   Department of Medical Oncology, Centre Georges-François Leclerc, Dijon, France
22   Departement de Médicine Oncologique, Gustave Roussy Cancer Campus, Villejuif, France
23   Département de Chirurgie, Gustave Roussy, Université Paris-Saclay, Villejuif, France
24   Service d'Urologie, Hôpital Foch, Suresnes, France
25   UVSQ – Université Paris Saclay, Versailles, France
26   Gustave Roussy, Université Paris-Saclay, Villejuif, France
27   AP-HP, Centre – Université de Paris, Hôpital Cochin, Unité d'Oncologie Thoracique, Service de Pneumologie, Paris, France
28   AP-HP, Hôpitaux Universitaires de l'Est Parisien, Hôpital Tenon, Service de Pneumologie, Paris, France
29   Service de Chirurgie Thoracique, Hôpital Marie Lannelongue, Le Plessis-Robinson, France
30   Team Inflammation, Complement and Cancer, INSERM, Centre de Recherche des Cordeliers, Paris, France
31   Sorbonne Université, Paris, France
32   Department of Pediatric and Adolescent Oncology, Gustave Roussy Cancer Campus, Université Paris-Saclay, Villejuif, France
33   Centre de Recherche des Cordeliers, Equipe labellisée par la Ligue contre le cancer, Université de Paris, Sorbonne Université, INSERM U1138, Institut Universitaire de France, Paris, France
34   Metabolomics and Cell Biology Platforms, Institut Gustave Roussy, Villejuif, France
35   Pôle de Biologie, Hôpital Européen Georges Pompidou, AP-HP, Paris, France
36   Suzhou Institute for Systems Medicine, Chinese Academy of Medical Sciences, Suzhou, China
37   Department of Women's and Children's Health, Karolinska Institute, Karolinska University Hospital, Stockholm, Sweden
38   Département d'Innovation Thérapeutique et d'Essais Précoces (DITEP), Gustave Roussy, Université Paris-Saclay, Villejuif, France
39   Center of Clinical Investigations in Biotherapies of Cancer (CICBT) 1428, Villejuif, France
     *Corresponding author: Tel: +33 14211655005; E-mail: laurence.zitvogel@gustaveroussy.fr
     †These authors contributed equally to this work as first authors

    

## Abstract

**Decision making in immuno-oncology is pivotal to adapt therapy to the tumor microenvironment (TME) of the patient among the numerous options of monoclonal antibodies or small molecules. Predicting the best combinatorial regimen remains an unmet medical need. Here, we report a multiplex functional and dynamic immuno-assay based on the capacity of the TME to respond to *ex vivo* stimulation with twelve immunomodulators including immune checkpoint inhibitors (ICI) in 43 human primary tumors. This "in sitro" (in situ/in vitro) assay has the potential to predict unresponsiveness to anti-PD-1 mAbs, and to detect the most appropriate and personalized combinatorial regimen. Prospective clinical trials are awaited to validate this in sitro assay.**

**Keywords** "in sitro" assay; cancer; immune checkpoint inhibitors; immunomonitoring; precision oncology
**Subject Categories** Cancer; Immunology

## Introduction

While immunotherapy has made great strides as a standalone and combined with conventional cytotoxic strategies, its effect is limited across tumor types and patient subsets (Topalian, 2015; Kalbasi & Ribas, 2020). The recent characterizations of multiple immune resistance mechanisms have fueled the development of novel agents to circumvent such limitations (Williams *et al*, 2020). Yet, the ability to predict and best overcome tumor resistance is not currently possible (Kalbasi & Ribas, 2020). A promising approach to circumvent primary resistance to programmed cell death-1 (PD-1) blockade is to target new immune inhibitory or agonistic checkpoints (Kalbasi & Ribas, 2020). In the near future, the use of combination strategies will increase the number of patients who are likely to benefit from immunotherapy (Riaz *et al*, 2017). However, several critical issues have yet to be addressed. First, the development and validation of predictive immune biomarkers are needed to guide immuno-oncology (I-O) treatment decisions across various malignancies. Secondly, the rapid diagnosis and rationale lending support to personalized combinatorial regimens will require a formalized scientific and clinical framework. Hence, it remains to be seen whether the future of I-O will rely on patient stratification or personalization.

Accumulating evidence highlight the capacity of anti-PD-1/PDL-1 monoclonal antibodies (mAbs) to target tumor infiltrating lymphocytes (TILs) *in situ* within tumor beds (Wei *et al*, 2018). Tertiary lymphoid organs (Thommen *et al*, 2018), temporal and metabolic changes in T-cell clones according to regional nonsynonymous tumor mutations (Inoue *et al*, 2016; Riaz *et al*, 2017; Joshi *et al*, 2019), and the efficacy differences between adjuvant and neoadjuvant settings (Liu *et al*, 2016) support the critical impact of the tumor microenvironment (TME) at the start of I-O to dictate the early outcome at 6–8 weeks first CT scan. Therefore, technical approaches directly assessing the dynamic functionality of immune checkpoint inhibitors on the native TME of accessible tumors may be instrumental to accelerate decision making in clinical management.

To do so, we developed a dynamic system biology approach aimed at defining key patient immunometrics. These metrics outlined relevant "*in situ*" prognostications of patient response through "*ex vivo*" reactivity of melanoma to cytotoxic T-lymphocyte-associated protein 4 (CTLA4) blockade (Jacquelot *et al*, 2017). In extending this high content screening to various histological types of primary tumors amenable to PD-1 blockade, we refined this *in sitro* assay using an immunoreactivity scoring of 17 selected soluble parameters to best assess the functional potential of immune infiltrates to twelve immunomodulators combined to anti-PD-1 mAbs. This *in vitro/in situ* ("*in sitro*") assay could identify surrogate markers of immune reactivity and had the potential to predict *in vivo* responses to anti-PD-1 mAbs.

## Results and Discussion

### Segregation of responding (R) and non-responding (NR) tumors based on *in sitro* reactivity to anti-PD-1 mAbs

Our population consisted of 43 patients with resectable and analyzable tumors (NSCLC [L, $n = 16$], kidney [K, $n = 20$], head and neck [HN, $n = 4$], ovarian [O, $n = 1$], and urothelial carcinoma [B, $n = 2$]) with 41/43 being naive untreated patients benefiting from a surgery in various Paris hospitals (refer to patients characteristics in Appendix Table S1). Among tumors > 2 cm in size ($n = \sim 120$) emanating from the operating room, we processed 43 tumor samples that met our internal eligible criteria for *in sitro* assays (cut-off value for alive $CD45^+$ cells: > 0.2% corresponding to 10–20 and 100–200/tumor infiltrated lymphocyte (TIL) $mm^2$ within tumor nests and stroma, respectively (Fig EV1A), and absolute cell number > 1 million; Appendix Tables S2 and S3). Of the tumor samples available for immunophenotyping using 90 parameter-based flow cytometry at D0 ($n = 34$), $CD45^-$ tumor/stromal cells represented $23.9 \pm 3.7\%$ of the TME. The phenotype of $CD45^+$ leukocytes comprising 10 cell types is comprehensively presented in Appendix Table S1–S3 and Appendix Figs S1 and S2.

We next analyzed the dynamics of the TILs within their native, although dissociated, TME after 3 days of *in sitro* stimulation with anti-PD-1 mAbs through the investigation of conventional effector lymphocyte functions, regulatory T cell (Treg) cells ($CD25^{hi}$-$Foxp3^+CD4^+$) and secretory patterns of the mixture (Fig 1A, Appendix Table S1–S3). Following stimulation with anti-PD-1 mAbs, and normalization onto medium values to classify tumor responsiveness (medium values being mostly equivalent to isotype control mAbs values [Appendix Fig S3]), we used a non-supervised hierarchical clustering of $z$ score-normalized concentrations of multiple ($n = 27$) immune and non-immune soluble factors (SFs) monitored by beads-based multiplex assay. The heatmap of this clustering highlights two categories of TME. Approximately, 17% (7/42) (4 L, 2 K, 1 HN) of tumors exhibited increased levels of most analytes above the mean of the whole cohort after stimulation with anti-PD-1 mAbs (called henceforth "$R_{clus}$", Fig 1B). The most significant differences between tumors responding ($R_{clus}$) or not responding ($NR_{clus}$) to anti-PD-1 mAbs resided in the release of CXCL10, GM-CSF, PDGF-bb, eotaxin, and IL-5; as well as inflammatory cytokines (IL-1$\beta$, tumor necrosis factor [TNF]$\alpha$, IL-17; Fig 1C) and not the usual Th1 cytokines. Most of these immunometrics were

compatible with the prominent soluble mediators secreted after T-cell receptor (TCR) cross-linking (Fig EV1B). In $R_{clus}$, the concentrations of the 27 cytokines/chemokines after 60 h of anti-PD-1 mAbs correlated with the percentages of $CD8^+$ T cells within $CD45^+$ leukocytes, as well as the effector functions of the immune infiltrate (Ki67, TNFα, interferon [IFN]γ, granzyme B [GrzB]; Fig EV1C, left panel). In $NR_{clus}$ tumors, however, the Th1 chemokine CXCL10 exhibited inverse correlation with the proportion of $CD8^+$ T cells (Fig EV1C, right panel). Of note, the integration of all 39 immunometrics did not improve the clustering. To refine the classifier, we attributed individual scores based exclusively on TCR-dependent SFs. They were defined as analytes for which the fold ratio between unstimulated and anti-CD3/anti-CD28 mAbs-stimulated tumors (concentration $_{anti-CD3/anti-CD28}$/concentration $_{medium}$) was superior to 1.5 (Fig EV1B, right panel). The "immune reactivity score" (IRS) assigned +1 to each of the 17 TCR-dependent SFs reaching $\geq$ 1.5-fold ratio following PD-1 blockade (concentration $_{anti-PD-1}$/concentration $_{medium}$) and integrated the sum of these 17 parameters (transformed in a percent value). Tumors accrual in IRS is depicted in Fig 1D and detailed in Appendix Table S4. "Immune reactive" status was defined by the cutoff of IRS $\geq$ 41.18 which was determined to maximize the sensibility and specificity for the best concordance with the hierarchical clustering classifier (Fig EV1D). This means that patients classified as $IRS_{high}$ have $\geq$ 7 out of 17 parameters that are increased by greater than or equal to 1.5-fold ratio. 7/42 (17%) of the tumor samples fell into this category including 5 L, 1 K, and 1 HN cancers. As expected, most of them corresponded to the specimens considered "$R_{clus}$" in the non-supervised hierarchical clustering method. Given the expected clinical objective response rate obtained with systemic administration of anti-PD-1 mAbs across malignancies, we surmised that the *in sitro* IRS may be a valuable tool to evaluate the likelihood of a patient to respond to PD-1 blockade. "*In vivo veritas*" could then be prospectively validated by assessing the definitive clinical response of six of our patients (L3, L6, L8, K1, K7, and K11) for seven *in sitro* assays and evaluations in the course of anti-PD-1 ± anti-CTLA4 mAbs administration for disease progression. In 5/7 assays, the *in sitro* IRS aligned with the clinical outcome (Fig 1E, Appendix Table S5). One patient enrolled in the TITAN study (K11) started with 8 weeks on anti-PD-1 treatment. Because of disease stability, K11 was switched to the combination of anti-PD-1 + anti-CTLA4. This patient demonstrated an improved response to combination checkpoint blockade following single line PD-1 blockade. For K11, the IRS was capable of predicting the response to both anti-PD-1 alone and combinatorial checkpoint blockade (detailed thereafter in Fig 3, Appendix Table S5). However, in one case where the IRS did not correspond to the clinical response (L8), we noticed that, following primary tumor resection, the patient benefited from local irradiation on a distant lesion concomitant to PD-1 blockade. It is possible that the local irradiation may have stimulated an abscopal response, suggesting that the IRS should have been evaluated post-irradiation in the lesion exposed to X-rays. Consequently, the *in sitro* platform analyzing 39 immunometrics allowed the *a priori* segregation of tumors in anti-PD-1 R or NR tumors using two converging methods taking into account 17 soluble analytes, that correlated with effector functions of TILs only in R tumors. Higher sample size is required to improve the statistical power of this prediction prospectively.

## Predictors of response or resistance to PD-1 blockade

We next considered SFs that are spontaneously released immediately after surgery contained in the transport medium prior to being processed in the laboratory, henceforth referred to as "tumor supernatants" (Fig 1A). No meaningful correlations were observed between SFs from tumor supernatants and their TIL phenotypes after PD-1 blockade (60 h; Fig 2A). We next analyzed potential links between such SFs and the IRS (< or > 41.18). Only one chemokine, the Th1-associated CXCL10, significantly correlated with the response to anti-PD-1 mAbs (IRS > 41.18; Fig 2B and C). Vascular endothelial growth factor tended to positively predict response to PD-1 blockade, possibly reflecting tumor lymphangiogenesis and tertiary lymphoid organogenesis (Fankhauser *et al*, 2017). In contrast, IL-8 tended to correlate with resistance to PD-1 blockade, as recently reported (Schalper *et al*, 2020). We next examined correlations between the IRS and the fluorescence-assisted cell sorting (FACS)-based TIL phenotypes at diagnosis (D0; Figs EV2 and EV3). This approach highlighted two potential predictors of response to PD-1 blockade: the expression of glucocorticoid-induced TNFR-related protein (GITR) on $CD3^+$ $CD56^+$ T cells (Figs 2D and EV2G) and the content in $CD3^-CD56^-$ cells (Figs 2D and EV2A). However, membrane PD-1 ligand (PD-L1) on $CD4^+$ $CD8^+$ T cells was associated with resistance to PD-1 blockade, as already reported (Jacquelot *et al*, 2017) (Fig 2D and EV3).

Finally, in order to validate the only significant marker that the *in sitro* platform highlighted, we retrospectively analyzed the predictive value of tumor CXCL10 expression (evaluated by RNA-seq on tumor biopsies at diagnosis) for the response to PD-1 blockade in two independent cohorts of L cancer patients ($N = 94$ in total) who had a follow-up > 6 months after PD-1 blockade, considering progression-free survival as a continuous variable or a cutoff at 6 months. Indeed, tumoral CXCL10 expression at diagnosis was significantly associated with prolonged PFS in univariate analysis (Appendix Table S6, Fig 2E and F).

Altogether, the *in sitro* immunodynamic analysis allowed us to re-enforce previously reported biomarkers, such as the good and bad prognostic value of CXCL10 (Choueiri *et al*, 2016), IL-8 (Schalper *et al*, 2020), and PDL-1 on double-positive $CD4^+$ $CD8^+$ T cells (Jacquelot *et al*, 2017; Menard *et al*, 2018), supporting the rationale of using this accelerated approach in clinical decision making.

## Precision Immuno-Oncology based on *in sitro* reactivity of immune infiltrates to I-O compounds

Next, we addressed whether this *in sitro* assay would reveal if tumors were susceptible to immune reactivity with PD-1 blockade alone or combined with other I-O compounds. Each of these compounds was used at the saturation dosing. Bearing in mind that sample number per combination test are different, out of seven $IRS_{high}$ samples (IRS > 41.18), 4 exhibited a slightly improved IRS adding either anti-CTLA4 (K11, Fig 3A–C), anti-CD73 (L16, Fig 3A and D), anti-GITR (K11, HN3, Fig 3A), or IDOi (L1, Fig 3A). Interestingly, the defucosylated Fc portion of anti-CTLA4 mAb, expected to increase antibody-dependent cell cytotoxicity and removal of Treg (Arce Vargas *et al*, 2018) did not improve the immune status of the tumors over the native compound (Fig 3A–C). When turning to tumors that failed to exhibit a response to anti-PD-1 mAbs *in sitro*

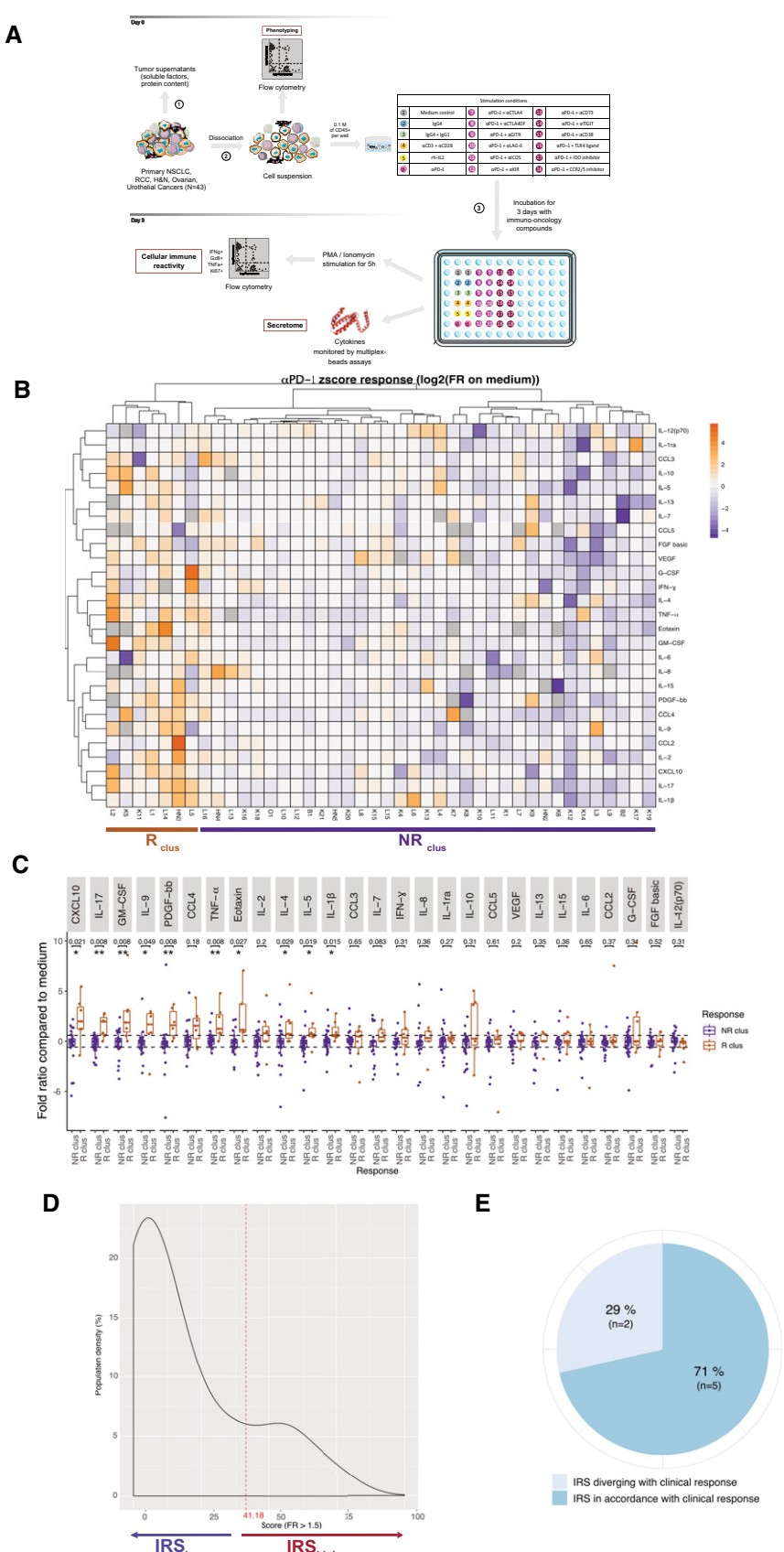

Figure 1.

**Figure 1. Clustering of the TME secretory pattern based on the *in sitro* platform to stratify response or resistance to PD-1 blockade.**

A  Overview of the *in sitro* diagnosis platform.

B  Heatmap of the non-supervised hierarchical clustering ($n = 42$) using 27 soluble factors (SF), secreted after 60 h-PD-1 blockade. Missing values are shown in gray. Both rows and columns are clustered using correlation distance and average linkage.

C  Fold ratios and ranges between $R_{clus}$ and $NR_{clus}$ for each SF. Box plots display group of numerical data through their $3^{rd}$ and $1^{st}$ quartiles (box), mean (central band), minimum and maximum (whiskers). Wilcoxon rank-sum test and P values with a Benjamini–Hochberg (BH) correction procedure (*$P < 0.05$, **$P < 0.01$) are indicated for each SF.

D  Density of patients according to the immune reactivity score (IRS), with threshold of positivity at 41.18 is indicated in dashed line.

E  *In vivo veritas* "validation" of the *in sitro* platform on seven available clinical data sets.

(IRS < 41.18), we first analyzed those that were also non-responders to recombinant IL-2 (rIL-2; $N = 19/27$; Appendix Table S7). Tumors anergic to rIL-2 exhibited higher basal expression of PD-1 or PDL-1 on T, natural killer (NK), B, and myeloid cells (Appendix Fig S4A). Among these, 26.3% (5/19) anergic TME could be rescued with an anti-PD-1 mAb-based combinatorial regimen. Interestingly, of the 8 tumors exhibiting an anti-PD-1 IRS < 41.18 but an rIL-2 IRS of ≥ 41.18, 6/7 responded to at least one combinatorial regimen and one could not be tested with other compounds. Considering all tumors harboring an IRS < 41.18, and tested for combinations specified, 50% (13/26) could be rescued by at least one anti-PD-1 mAb-based combination (Fig 3E). Not all combinations were tested in the 26 tumors (refer to Fig 3E). In a rare case (K18), combination with any tested compounds, except anti-CTLA4, was effective (Fig 3F and G, left panel), while L12 (Fig 3F and G, right panel) required the addition of Toll-like receptor 4 (TLR4) agonist. There was no correlation in our assay between the cell surface expression of the target molecules tested in tumors at baseline and the immune reactivity to each targeting mAbs (IRS) and any respective combinatorial regimen (Appendix Fig S4).

Altogether, the *in sitro* assay indicated that 50% of IRS$_{low}$ to anti-PD-1 mAbs could be rescued by a combination. However, optimal responses harbored a variegated profile, suggesting that personalization as opposed to stratification may in fact be the preferred solution to circumvent primary resistance to PD-1 blockade (Fig 3E, lower line). Additional studies to increase the sample size for each combination are needed for robust determination of therapeutic complementarity in this system.

### Hypo-responsiveness induced by PD-1 blockade

Cancer hyper-progression during the first four courses of systemic anti-PD-1 mAbs remains a potential concern (Champiat *et al*, 2018). Intrigued by a drop in cytokine release profile in some tumors (Fig 1B), not explained by activation-induced cell death of lymphocytes or increased proliferation of tumor cells, we designed a score of hyporeactivity for the 27 SFs of the multiplex array, meaning a diminution by 10 of the fold ratio (FR < 0.1) between anti-PD-1 stimulated vs non stimulated TILs, affecting +1 for each SFs and the sum of them for the hypo-responsive score (HRS). The median of the r was 5.55 (Fig EV4A and B) and 6 tumors presented a HRS > 5.55 (Appendix Table S8). The only predictive factor associated with HRS$_{high}$ was the co-expression of CTLA4 and C-C chemokine receptor 5 (CCR5) in tumor Treg (Fig EV4C and D), as already described (Kamada *et al*, 2019). Elimination of Tregs by FACS-cell sorting allowed for increased cytokine release, while adding them back into the coculture prevented it (Fig EV4E). Moreover, anti-Killer cell immunoglobulin-like receptor (KIR) mAbs was the best condition to prevent anti-PD-1 mAbs-induced HRS > 5.55 (Fig EV4F, Appendix Table S8). Thus, the *in sitro* platform may be able to detect a hypo-responsive TME, based on three criteria: (i) an elevated HRS (> 5.55) following PD-1 blockade, (ii) the presence of a particular Treg subset, and (iii) the efficacy of anti-KIR mAb in reducing the HRS under its threshold. This interesting result deserves prospective validation in clinical trials.

Finally, our initial attempt to scale down the 17 plex-based *in sitro* platform, from a whole tumor to a large biopsy, was not a major technical challenge. Hence, prospective clinical trials are warranted to validate these results to develop a platform as decision aid tool for future I.O.

## Material and Methods

### Patients and cohorts characteristics

#### Prospective cohort of 43 patients

Patients over 18 years old from Gustave Roussy Cancer Campus, Marie Lannelongue, Cochin, Tenon, Foch, Kremlin-Bicêtre, and Saint Joseph hospitals, with stage I/II/III lung (L, $n = 16$), kidney (K, $n = 20$), head and neck (HN, $n = 4$), ovarian (O, $n = 1$), and

**Figure 2. Biomarkers associated with response to *in sitro* PD-1 blockade.**

A  Spearman correlation matrix between concentrations of SF in tumor supernatants and FACS-based immune effector functions at 60 h post-PD-1 blockade (in fold ratio over medium alone) for $n = 43$ tumors. *$P < 0.05$.

B–D  Volcano plot of IRS (IRS > 41.2 underlined in red) and the concentrations of SF for 43 tumors (B) or phenotype of TILs (D) at baseline (CXCL10; $P = 0.055$) of log$_{10}$-transformed Wilcoxon rank-sum test P values with a Benjamini–Hochberg (BH) correction procedure (*FDR*) or not (*P*) and the log$_{10}$-transformed ratio according to the IRS. Significant biomarkers are filled in red (FDR < 0.2) and associated biomarkers ($P < 0.051$) are highlighted. (C) Detailed bar graph of the concentrations of SF in (IRS$_{high}$ > 41.2, $n = 4$) and (IRS$_{low}$, $n = 18$) tumors. Box plots display group of numerical data through their $3^{rd}$ and $1^{st}$ quartiles (box), mean (central band), minimum and maximum (whiskers). Statistical analyses: Wilcoxon rank-sum test, ns. $P > 0.05$, *$P < 0.05$.

E, F  *In vivo* validation of the predictive value of tumor CXCL10 for the response to PD-1 blockade in NSCLC patients. Kaplan–Meier progression-free survival (PFS) curves according to the median value (E) or cutoff value (< 6 months, $n = 61$; > 6 months, $n = 32$) (F) of tumor CXCL10 relative expression from two independent cohorts of NSCLC patients. (Appendix Table S6).

Source data are available online for this figure.

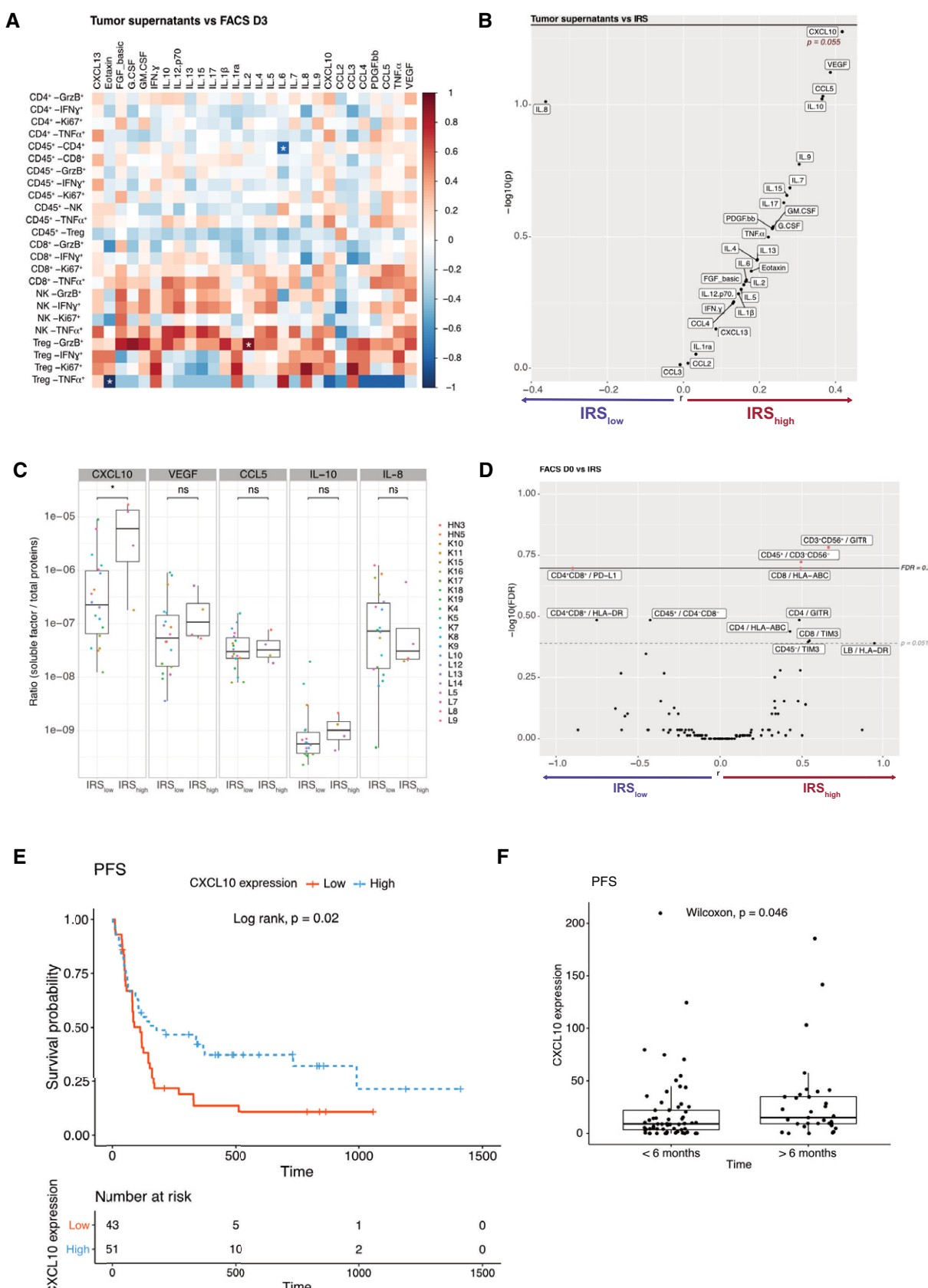

**Figure 2.**

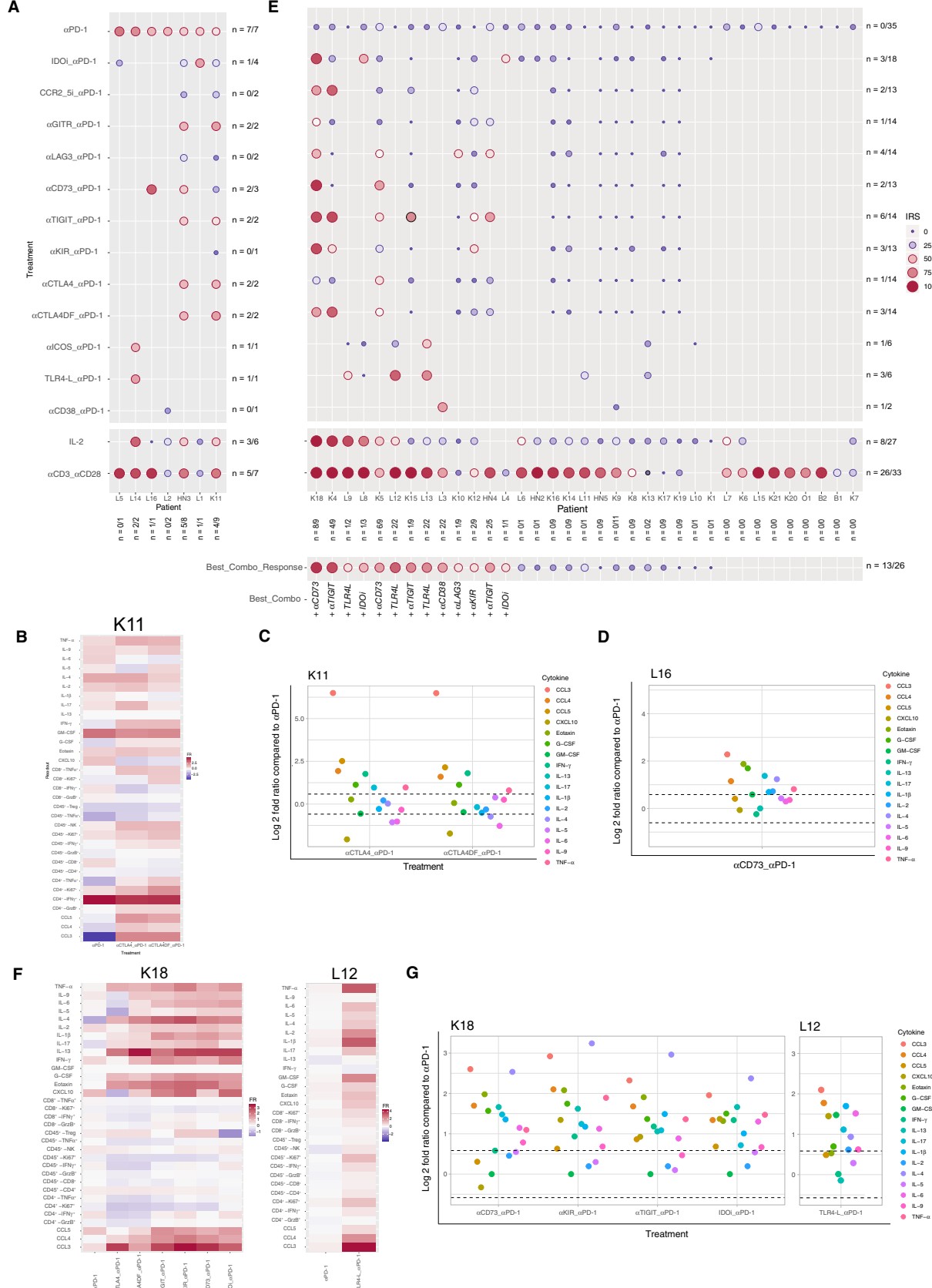

**Figure 3.**

**Figure 3.  Reactivating anergic TME with anti-PD-1 mAb-based combinatorial regimen.**

A–G  (A, E) The IRS was determined for each tumor after PD-1 blockade alone (top line), or combined with immunomodulators in two groups of tumors (IRS$_{anti-PD-1}$> (A) or < 41.18 (E)). The size and color of the bullet both correspond to the IRS. The "*n*" corresponds to the number of immune reactive tumors for each combination (horizontally) and to the number of combinations that induced a positive immune reactivity ($\geq$ 41.18) per tumor (vertically). Best corresponding combination responses are indicated below for each patient. Note that these experiments do not allow to establish direct comparisons of relative efficacy in-between each compound, given the limited amount of samples tested, the lack of dose ranges tested for each compound and/or sample availability. (B–D, F, G) Focus on four cases for the reactivity to anti-PD-1 mAbs alone or combined with immune checkpoints. Heatmap depicting the fold ratio for each immunometrics between stimulation with mAbs vs medium and raw data illustrating the increase of the soluble factors (SFs) of the combinations compared to anti-PD-1 alone.

bladder carcinoma (B, $n = 2$) primary resectable tumors provided written informed consent according with protocols reviewed and approved by institutional ethics committee including the investigator-sponsored, study "mAb *in sitro* test", N°ID-RCB: 2016-A00732-49. The experiments conformed to the principles set out in the WMA Declaration of Helsinki and the Department of Health and Human Services Belmont Report.

## Immunohistochemistry

4 μm-thick sections were prepared from tissue samples fixed in buffered formalin pH 7.4 (Merck, Darmstadt, Germany), embedded in paraffin wax (Sakura, Alphen, The Netherlands), and cut with a microtome (Leica, Wetzlar, Germany). One section was stained with hematoxylin–eosin and used for the evaluation of TILs, which was performed according to international recommendations (Hendry et al, 2017). One section was used for CD45 immunodetection. Briefly, an indirect immunoperoxidase technique was applied to deparaffinized sections. A combination of two CD45 mouse mAbs, diluted 1:200, was used (2B11 + P7/26; Dako, Glostrup, DK). The reaction was performed with an automated stainer (Bond RX, Leica, Wetzlar, Germany). The density of CD45$^+$ cells was evaluated within the tumor tissue by a senior pathologist; labeled cells were counted visually in at least 10 high power fields of 1 mm$^2$ each; the result was expressed as the mean cell number/mm$^2$.

## Tumor supernatants from resected cancer specimens

Resected pieces were kept in NaCl or RPMI 1640 (GIBCO Life Technologies, ref: 31870-025) for 6–18 h prior to dissociation. After 5 min at 425 *g* centrifugation, supernatants, called "tumor supernatants", were frozen at −80°C until their use. These supernatants were controlled and normalized for total proteins concentration. To do this protein calibration, 1 μl of tumor supernatants was tested using Micro BCA™ Protein Assay Kit (Thermo Fisher, ref: 23235) following manufacturer's protocol; and 50 μl of tumor supernatants were used for bead-based multiplex immunoassays (Luminex™ technology) following manufacturer's instructions. SFs were then normalized by the total proteins concentration.

## Tumor infiltrated lymphocyte preparations

Resected cancer specimens from 43 patients were cut and placed in dissociation medium, which consisted of RPMI 1640 (GIBCO Life Technologies, ref: 31870-025), Collagenase IV (50 IU/ml; Sigma-Aldrich, ref: C2139), Hyaluronidase (280 IU/ml; Sigma-Aldrich, ref: H6254), and DNAse I (30 IU/ml; Sigma-Aldrich, ref: 260913), and run on a gentle MACS OctoDissociator (Miltenyi Biotec).

Dissociation time lasted 1 h under mechanical rotation and heating. Cell samples were diluted in PBS, passed through a cell strainer, and centrifuged for 5 min at 1,500 rpm. Cells were finally resuspended in PBS, counted with Precision Count Beads™ (Biolegend, ref: 424902) by flow cytometry following manufacturer's protocol, then stained for flow cytometric analyses or resuspended in fetal calf serum containing 10% of dimethyl sulfoxide (DMSO; SIGMA, ref: 276855) for storage in liquid nitrogen.

## *Ex vivo* tumor assays (depicted in Fig 1A)

Dissociated cells from K, L, HN, O, B tumors (66% freshly harvested and 34% from frozen in DMSO) were stained for D0 phenotyping ($0.5-0.25 \times 10^6$ of CD45$^+$ cells per panel) and incubated in 96-well plate at $0.1 \times 10^6$ of CD45$^+$ cells per well in complete medium (RPMI 1640 (GIBCO Life Technologies, ref: 31870-025) supplemented with 10% human AB serum (Institute Jacques Boy, ref: 201021334), 1% Penicillin/Streptomycin (GIBCO Invitrogen, ref:15140-122), 1% L-glutamine (GIBCO Life Technologies, ref: 25030-024) and 1% of sodium pyruvate (GIBCO Life Technologies, ref: 11360-039)) and with isotype control, agonist or antagonist mAbs, or with small molecules as described in Fig 1A. After 60 h of incubation with or without drugs, supernatants were collected for bead-based multiplex immunoassays (Luminex™ technology) and cells were stimulated with PMA (5 ng/ml; Sigma-Aldrich, ref: 524400), ionomycin (125 ng/ml; Sigma, ref: 10634), brefeldin A (1 μl/ml; eBioscience, ref: 00-4506-51), and Golgi Stop (4 μl/6 ml; BD Biosciences, ref: 554724). After 5 h, cells were harvested and then labeled for membranous and intracellular molecules according to the manufacturer's protocol.

## Immune checkpoint inhibitors and immunostimulatory reagents

The cell suspension was stimulated for 60 h with the following reagents: anti-PD-1 (nivolumab [Bristol-Myers Squibb (BMS)] and pembrolizumab [Merck], 10 μg/ml), isotypes (IgG4 or IgG1, 10 μg/ml), anti-CD38 (daratumumab, Janssen-Cilag, IgG1, 5 μg/ml), anti-CD73 (BMS-986179, BMS, IgG1, 10 μg/ml), anti-CTLA4 (ipilimumab, BMS, IgG1, 10 μg/ml), anti-CTLA4 defucosylated (CTLA4DF, BMS, IgG1, 10 μg/ml), anti-GITR (BMS-986156, BMS, IgG1, 10 μg/ml), anti-inducible T-cell co-stimulator (GSK3359609, IgG4, GlaxoSmithKline (GSK), 10 μg/ml), anti-KIR (BMS-986015, BMS, IgG4, 10 μg/ml), anti-lymphocyte-activation gene 3 (LAG3; BMS-986016, BMS, IgG4, 10 μg/ml), anti-T-cell immunoreceptor with Ig and ITIM domains (TIGIT; BMS-986207, BMS, IgG1, 10 μg/ml), CCR2/CCR5 inhibitor (BMS-6876814, BMS, 10 nM), indolamine 2,3-dioxygenase inhibitor (INCB024360, Incyte, 5 mM; BMS-986205, BMS, 125 ng/ml), TLR4 agonist (GSK1795091A, 10 μg/ml), anti-CD3

(Thermo Fisher Scientific, clone OKT3, 10 μg/ml), anti-CD28 (Thermo Fisher Scientific, clone CD28.2, 10 μg/ml), and rIL-2 (Pepro-Tech, ref: 200-02-11, 10 μg/ml). Each molecule was used at the saturating dosing according to manufacturer's recommendations.

## Flow cytometric analyses

For membranous labeling, TILs were stained to discriminate different lymphocyte subsets with fluorochrome-coupled mAbs incubated for 15 min at room temperature (RT) and washed. Intracellular staining was performed after permeabilization with forkhead box P3 (FoxP3)/ Transcription Factor Staining Buffer Set (Thermo Fisher Scientific, ref: 00-5523-00) and intracellularly labeled with anti-Foxp3-APC (eBiosciences, clone PCH101) mAb at D0, or with anti- IFN-γ-PECy7 (BD Biosciences, clone 4S.B3), anti-TNFα-BV650 (BD Biosciences, clone MAb11), anti-GrzB-PECF594 (BD Biosciences, clone GB11) mAbs and Ki67 (BioLegend, ref: 350514 or BD Biosciences, ref: 556027) at D3, following the manufacturer's protocol. Cell samples were acquired on a BD LSRFortessa™ X-20 flow cytometer (BD Biosciences) with single-stained antibody-capturing beads used for compensation (CompBeads, BD Biosciences, ref: 552843). Data were analyzed with Kaluza Analysis software v2.1 (Beckman Coulter). Of the tumor samples available for flow cytometry at D0 ($n = 34$), $CD45^-$ tumor/stromal cells represented $23.9 \pm 3.7\%$ of the TME. The $CD45^+$ leukocyte composition, including T, B, NK, and myeloid cells were analyzed by flow cytometry in available specimens (Appendix Table S1–S3, Appendix Figs S1 and S2A and B). Based on the immunophenotyping of 20–90 parameters (10 cell types and 2–9 mAbs target molecules per patient), we found comparable and variable distributions of T, NK, B and myeloid cells across individuals and tumor types (Appendix Fig S2B). However, K carcinomas were the only cancer subtype we tested that presented more than 5% of $CD4^+ CD8^+$ cells (Appendix Fig S2B). Those $CD4^+CD8^+$ cells, rather $CD4^{dim}CD8^{bright}$, tended to express higher levels of human leukocyte antigen (HLA)-DR, PD-1, CD73, and T-cell immunoglobulin and mucin domain-containing protein 3 (TIM-3) than their single positive $CD4^+$ T or $CD8^+$ T counterpart cells (Appendix Fig S2D, E, G and H), in accordance with previous work (Menard et al, 2018). Interestingly, while TILs from K tended to express more TIGIT than HN tumors, they also harbored significantly less CD73 on $CD3^-CD56^-$ and NK cells ($P < 0.05$, $P = 0.14$, respectively), while presenting less HLA-DR compared with HN cancers (Appendix Fig S2G–J). As expected, HLA-ABC was less expressed in tumor cells than leukocytes (Appendix Fig S2C). Concurrently, $CD4^+$ T, $CD8^+$ T, and NK cells from L cancers seemed to express higher levels of GITR than in the other tumor types (Appendix Fig 2I). Of note, PD-L1 was not broadly expressed on $CD45^+$ or $CD45^-$ TME across these tumor types (Appendix Fig S2F).

## Cytokines and chemokines measurements

Tumor supernatants were monitored using the Bio-Plex Pro™ Human Chemokine BCA-1/chemokine (C-X-C motif) ligand 13 set (Bio-Rad, ref: 171BK12MR2) and the Bio-Plex Pro™ Human Cytokine 27-plex Assay (Bio-Rad, ref: M500KCAF0Y). Supernatants from cultured cells at D3 were monitored using the Bio-Plex Pro™ Human Cytokine 27-plex Assay (Bio-Rad, ref: M500KCAF0Y) according to

the manufacturer's instructions. Acquisitions and analyses were performed on a Bio-Plex 200 system (Bio-Rad) and a Bio-Plex Manager 6.1 Software (Bio-Rad), respectively.

## Scores

The immune reactivity score (IRS) was calculated taking into account the 17 TCR-dependent SFs (those reaching a median of the fold ratio following TCR cross-linking [concentration $_{aCD3/aCD28}$/concentration $_{medium}$] > 1.5 [refer to Fig EV1A]). We assigned +1 to each TCR-dependent SF reaching > 1.5-fold ratio after stimulation with anti-PD-1 mAbs (concentration $_{anti-PD-1}$/concentration $_{medium}$). The IRS corresponds to the sum of the positive SFs transformed in a percent value. A tumor was considered "immune reactive" (or $IRS_{high}$) when the immune reactivity score IRS ≥ 41.2. Table S3 summarizes each IRS for each patient tumor. The HRS was calculated taking into account all of the 27 SFs available. We assigned +1 to each SF reaching < 0.1-fold ratio after stimulation with anti-PD-1 mAbs (concentration $_{anti-PD-1}$/concentration$_{medium}$). The HRS corresponds to the sum of all 27 parameters, establishing a median at 5.5 defining the $HRS_{high}$ vs $HRS_{low}$. Figure EV4A summarizes each HRS for each patient tumor. Similar data were obtained using an isotype control mAb instead of medium (Appendix Fig S2).

## General statistical analysis

Data representations were performed either with Prism 6 (GraphPad San Diego, CA, USA) or R v3.6 using tidyverse, dplyr, ggplot2, ggpubr, pheatmap, corrplot, ggdendro, Hmisc, or survminer packages. Box plots display a group of numerical data through their $3^{rd}$ and $1^{st}$ quartiles (box), mean (central band), minimum and maximum (whiskers), and each dot represents one tumor sample. All calculations and statistical tests were performed using R v3.6. Unless stated, $P$ values are two-sided and 95% confidence intervals for the reported statistic of interest. Individual data points representing the measurement from one tumor are systematically calculated from the corresponding distribution. Wilcoxon rank-sum test was applied to assess differences in concentration between two different responses to treatment (R vs NR) or between different cell subtypes. When indicated, the false discovery rate (FDR, $P > 0.2$) was controlled using the Benjamini–Hochberg procedure. For the progression-free survival (PFS) analysis, Cox regression model using survival R package was used to estimate hazard ratio (HR) of the explanatory variables and their 95% confidence intervals (CIs). Survival curve was estimated by Kaplan–Meier method, using the R package survminer. Optimal cutoff for CXCL10 expression was chosen based on a maximally selected rank statistics (Kamada et al, 2019). SFs and flow cytometry parameters fold ratios were calculated as $log_2$ transformation of median values of stimulated vs unstimulated wells and were converted to $z$ scores. Figure 1B has been generated with the R package Pheatmap. Hierarchical clustering of the 42 patients based on the $z$ score of 27 SFs was performed using Euclidean distance and ward.D clustering. In Fig EV1C spearman correlation of SFs fold ratio (stimulated over unstimulated) clustering is performed using Euclidean distance and ward.D clustering. Spearman correlations are computed with the Hmisc R package. All $P$ values and number of each groups are depicted in Appendix $P$ value tables.

### The paper explained

#### Problem
Predicting primary resistance to PD-1 blockade and adapting combinatorial regimens in a stratified or personalized manner remain problematic in cancer treatment. Diagnosis tools for decision making are urgently needed in cancer patients.

#### Results
We set up a functional dynamic multiplexed immunophenotyping assay, measuring up to 50 parameters after 60 h of *ex vivo* stimulation with 12 immunomodulators. This flow cytometry- and Luminex-based assay was performed in 43 fresh tumors. We selected a 17 analyte-based immune reactivity score that detected primary resistance to PD-1 blockade. 26% of tumors exhibited both anergic tumor infiltrating lymphocytes (TILs) to rIL-2 and no responses to PD-1 blockade. The lack of CXCL10 release from fresh neoplasia was associated with primary resistance to PD-1 blockade. In seven cases amenable to *in vivo* therapy with PD-1 blockade, this "*in sitro*" assay anticipated the resistance to therapy in five tumors. 50% of primary resistance to PD-1 blockade could be shifted toward immunoreactivity using personalized combinatorial regimens. The presence of a specific regulatory T-cell subset correlated with anti-PD-1 antibody-induced immunosuppression that could be rescued by anti-killer inhibitory receptor mAbs.

#### Impact
Albeit exemplified in only 43 cases in this report, the *in sitro* assay has the power to predict primary resistance to PD-1 blockade and can propose a personalized combinatorial regimen of immunomodulators in 60 h based on freshly dissociated tumor specimen and a 17 analyte-based score, advocating for its future validation in prospective clinical trials.

## Data availability

This study includes no data deposited in external repositories.

**Expanded View** for this article is available online.

## Acknowledgements
LZ and GK were supported by the Ligue contre le Cancer (équipe labelisée); Agence Nationale de la Recherche (ANR) francogermanique ANR-19-CE15-0029, ANR Projects blancs; ANR under the frame of E-Rare-2, the ERA-Net for Research on Rare Diseases; Association pour la recherche sur le cancer (ARC); Bristol-Myers Squibb Company (International Immuno-Oncology Network), Cancéropôle Ile-de-France; Chancellerie des universités de Paris (Legs Poix), Fondation pour la Recherche Médicale (FRM); a donation by Elior; the European Commission (ArtForce); the European Research Council (ERC); Fondation Carrefour; Institut National du Cancer (INCa); Inserm (HTE); Institut Universitaire de France; LeDucq Foundation; the LabEx Immuno-Oncology; the RHU Torino Lumière (ANR-16-RHUS-0008); H2020 ONCO-BIOME, the Seerave Foundation; the SIRIC Stratified Oncology Cell DNA Repair and Tumor Immune Elimination (SOCRATE); FHU CARE, Dassault, and Badinter Philantropia, and the Paris Alliance of Cancer Research Institutes (PACRI). AC is supported by the CPRIT Research Training Program (RP170067). JEF was supported by Transgene and AGG was supported by FRM. AM has been or is currently an investigator in clinical trials sponsored by BMS, MSD, GSK/Tesaro, Janssen, Roche/Genentech, Pfizer, Astra Zeneca (AZ), Amgen. AM has been or is currently a member of Clinical Trial Scientific Steering Committee for AZ and GSK. AM has been or is currently a member of the scientific advisory board of the following companies: Merck Serono, Novartis, BMS, Symphogen, Amgen, Tesaro/GSK, Pfizer, Astra Zeneca/Medimmune, Servier, Sanofi. AM has provided Scientific & Medical Consulting to the following companies: Roche, Sanofi. AM is a member of the Data Safety and Monitoring Board for the following trial NCT02423863 (TLR3 agonist; Oncovir). AM has received research funding and or drug supply for pre-Clinical and clinical research projects from: BMS, Boehringer Ingelheim, Idera, MSD, Fondation MSD Avenir, SIRIC (INCa-DGOS-Inserm_12551).

## Author contributions
LZ designed and conducted the study, MB coordinated the ethical and experimental parts of the *in sitro* assay and interpreted data related to this work, AD, J-EF, and A-GG analyzed and interpreted data related to this work, AD, NV, CB, SL performed the *in sitro* experiments, NJ helped in the conception of the platform, J-Y Scoazec performed IHC evaluation, DD and LR contributed to the statistical analyses, DB, SM and SS provided experimental and scientific help, AC provided scientific help, EB performed the CXCL10 RNA-seq, BR and FG recruited patients for the CXCL10 RNA-seq analysis, ST and LD collected clinical information, DJF, SY, JDW, MB, AH and TC provided I-O compounds and scientific input, VTM, OC, CR, SF, BP, CG, IB, MR, CL, HB, MW, EF, IC, LA, YL helped in the recruitment and collection of the tumor specimen, FA, BG, J-YS, GK and AM provided scientific input.

## Conflicts of interest
LZ received bench fees and signed research contracts with Tusk Therapeutics, GSK, BMS, Incyte and Transgene to screen their pipeline which helped performing this extensive study over the past 5 years. GSK authors are employees and stockholders of GSK. Other authors declare no conflict of interest.

## For more information
UMR 1015 Immunologie des tumeurs et immunothérapie contre le cancer: https://www.gustaveroussy.fr/fr/umr1015

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
