## [Review Process File · EMBO Molecular Medicine]

Immunodynamics of Explanted Human Tumors for Immuno-Oncology

Agathe Dubuisson, Jean-Eudes Fahrner, Anne-Gaëlle Goubet, Safae Terrisse, Nicolas Voisin, Charles Bayard, Sebastien Lofek, Damien Drubay, Delphine Bredel, Séverine Mouraud, Sandrine Susini, Alexandria Cogdill, Lucas Rebuffet, Elise Ballot, Nicolas Jacquelot, Vincent Thomas de Montpreville, Odile Casiraghi, Camélia Radulescu, Sophie Ferlicot, David J. Figueroa, Sapna Yadavilli, Jeremy D. Waight, Marc Ballas, Axel Hoos, Thomas Condamine, Bastien Parier, Christophe Gaudillat, Bertrand Routy, François Ghiringhelli, Lisa Derosa, Ingrid Breuskin, Mathieu Rouanne, Fabrice André, Cédric Lebacle, Hervé Baumert, Marie Wislez, Elie Fadel, Isabelle Cremer, Laurence Albiges, Birgit Georger, Jean-Yves Scoazec, Yohann Lorient, Guido Kroemer, Aurélien Marabelle, Mélodie Bonvalet, Laurence Zitvogel **DOI: 10.15252/emmm.202012850**

Corresponding authors: Laurence Zitvogel (laurence.zitvogel@gustaveroussy.fr)

Review Timeline:

Submission Date:	30th May 20
Editorial Decision:	9th Jul 20
Revision Received:	29th Sep 20
Editorial Decision:	19th Oct 20
Revision Received:	26th Oct 20
Accepted:	28th Oct 20

Editor: Lise Roth

Transaction Report:

9th Jul 2020

Dear Prof. Zitvogel,

Thank you for the submission of your manuscript to EMBO Molecular Medicine. We have now received feedback from the three reviewers who agreed to evaluate your manuscript. As you will see from the reports below, the referees acknowledge the interest of the study and its potential clinical impact, and are overall supporting publication of your work pending appropriate revisions.

However, a major criticism shared by the referees is the small number of samples, which limits the predictive value of the assay. Therefore, more samples should be included in the study. Alternatively, the referees agree that the assay by itself is of interest, and would agree for the data to be published as a shorter report should you not be able to provide more samples. In that case, the claims should be toned down accordingly.

Addressing the other reviewers' concerns in full will be necessary for further considering the manuscript in our journal, and acceptance of the manuscript will entail a second round of review. EMBO Molecular Medicine encourages a single round of revision only and therefore, acceptance or rejection of the manuscript will depend on the completeness of your responses included in the next, final version of the manuscript. For this reason, and to save you from any frustrations in the end, I would strongly advise against returning an incomplete revision.

When submitting your revised manuscript, please carefully review the instructions that follow below. Failure to include requested items will delay the evaluation of your revision:

- 1) A .docx formatted version of the manuscript text (including legends for main figures, EV figures and tables). Please make sure that the changes are highlighted to be clearly visible.
- 2) Individual production quality figure files as .eps, .tif, .jpg (one file per figure).
- 3) A .docx formatted letter INCLUDING the reviewers' reports and your detailed point-by-point responses to their comments. As part of the EMBO Press transparent editorial process, the point-by-point response is part of the Review Process File (RPF), which will be published alongside your paper.
- 4) A complete author checklist, which you can download from our author guidelines (<https://www.embopress.org/page/journal/17574684/authorguide#submissionofrevisions>). Please insert information in the checklist that is also reflected in the manuscript. The completed author checklist will also be part of the RPF.
- 5) Please note that all corresponding authors are required to supply an ORCID ID for their name upon submission of a revised manuscript.
- 6) Before submitting your revision, primary datasets produced in this study need to be deposited in

an appropriate public database (see <https://www.embopress.org/page/journal/17574684/authorguide#dataavailability>). Please remember to provide a reviewer password if the datasets are not yet public. The accession numbers and database should be listed in a formal "Data Availability " section (placed after Materials & Method). Please note that the Data Availability Section is restricted to new primary data that are part of this study.

7) We would also encourage you to include the source data for figure panels that show essential data. Numerical data should be provided as individual .xls or .csv files (including a tab describing the data). For blots or microscopy, uncropped images should be submitted (using a zip archive if multiple images need to be supplied for one panel). Additional information on source data and instruction on how to label the files are available at .

8) Our journal encourages inclusion of *data citations in the reference list* to directly cite datasets that were re-used and obtained from public databases. Data citations in the article text are distinct from normal bibliographical citations and should directly link to the database records from which the data can be accessed. In the main text, data citations are formatted as follows: "Data ref: Smith et al, 2001" or "Data ref: NCBI Sequence Read Archive PRJNA342805, 2017". In the Reference list, data citations must be labeled with "[DATASET]". A data reference must provide the database name, accession number/identifiers and a resolvable link to the landing page from which the data can be accessed at the end of the reference. Further instructions are available at .

9) We replaced Supplementary Information with Expanded View (EV) Figures and Tables that are collapsible/expandable online. A maximum of 5 EV Figures can be typeset. EV Figures should be cited as 'Figure EV1, Figure EV2" etc... in the text and their respective legends should be included in the main text after the legends of regular figures.

- Additional Tables/Datasets should be labeled and referred to as Table EV1, Dataset EV1, etc. Legends have to be provided in a separate tab in case of .xls files. Alternatively, the legend can be supplied as a separate text file (README) and zipped together with the Table/Dataset file. See detailed instructions here:

10) The paper explained: EMBO Molecular Medicine articles are accompanied by a summary of the articles to emphasize the major findings in the paper and their medical implications for the non-specialist reader. Please provide a draft summary of your article highlighting

11) For more information: There is space at the end of each article to list relevant web links for further consultation by our readers. Could you identify some relevant ones and provide such information as well? Some examples are patient associations, relevant databases, OMIM/proteins/genes links, author's websites, etc...

12) Every published paper now includes a 'Synopsis' to further enhance discoverability. Synopses are displayed on the journal webpage and are freely accessible to all readers. They include a short stand first (maximum of 300 characters, including space) as well as 2-5 one-sentences bullet points that summarizes the paper. Please write the bullet points to summarize the key NEW findings. They should be designed to be complementary to the abstract - i.e. not repeat the same text. We encourage inclusion of key acronyms and quantitative information (maximum of 30 words / bullet point). Please use the passive voice. Please attach these in a separate file or send them by email, we will incorporate them accordingly.

Please also suggest a striking image or visual abstract to illustrate your article. If you do please provide a png file 550 px-wide x 400-px high.

13) As part of the EMBO Publications transparent editorial process initiative (see our Editorial at <http://embomolmed.embopress.org/content/2/9/329>), EMBO Molecular Medicine will publish online a Review Process File (RPF) to accompany accepted manuscripts.

In the event of acceptance, this file will be published in conjunction with your paper and will include the anonymous referee reports, your point-by-point response and all pertinent correspondence relating to the manuscript. Let us know whether you agree with the publication of the RPF and as here, if you want to remove or not any figures from it prior to publication.

I look forward to receiving your revised manuscript.

Yours sincerely,

Lise Roth

Lise Roth, PhD
Editor
EMBO Molecular Medicine

To submit your manuscript, please follow this link:

Link Not Available

Photos 400-800 DPI

*Additional important information regarding figures and illustrations can be found at <http://bit.ly/EMBOPressFigurePreparationGuideline>

***** Reviewer's comments *****

Referee #1 (Comments on Novelty/Model System for Author):

Well designed and conducted study, using human samples

Referee #1 (Remarks for Author):

Dubuisson et al. established an "in vitro" (ex vivo/in situ) assay to analyze the immune reactivity of cancer infiltrates within primary tumors to various immune checkpoint inhibitors and activators. The approach looks very interesting. However, the authors need to explain in more detail the experimental set-up (dynamic multiplexed immunophenotyping assay).

1. It is recommended to do a parameter study to show the calibration and robustness of the method. For example, they stated "A tumor was considered immune reactive (or IRShigh) when the immune reactivity score IRS {greater than or equal to} 41.2". is there any reference for this exact value?
2. Tumor supernatants were produced by both tumor cells and TILs. How did the authors ensure that the tumor samples were all of the same size or same weight? Please provide the size or weight of each sample.
3. Why did the incubation period range from 2 to 24 hours? Does incubation period affect the cytokine concentration?
4. The clinical data (S-table4) is very limited. This should be acknowledged and discussed as a limitation.

Referee #2 (Remarks for Author):

Immunodynamics of Explanted Human Tumors for Precision and

Personalized Immuno-Oncology

Dubuisson et al., developed a new immunoassay to guide treatment decision and personalized therapy in immuno-oncology. This assay is based on single cell suspensions from tumor samples and includes immune cell phenotyping (focus on lymphocytes) and measurement of soluble factors released by stimulated immune cells. Based on the assay parameters the authors defined an immune reactivity score used to distinguish tumors responsive to anti-PD1 therapy from non-responsive tumors (total n=43, 5 cancer entities). Applying this assay the authors could predict in 5/7 cases the clinical outcome of patients treated with PD-1 blockade. Moreover, the assay was used for identification of PD-1-based combinatorial regimens that shifted tumors with primary PD-1 resistance towards immuno-reactivity.

Reviewer:

With the new immunoassay and immune reactivity score the authors developed a broadly applicable platform to analyze tumor responses to single and combined anti-PD1-based treatments in vitro and guide therapy decisions in immuno-oncology. Data from a limited sample set (n=7) suggest that this assay might be suitable also in predicting response to anti-PD1 therapy. Overall, this assay format will strongly improve translational studies in immuno-oncology. Its predictive value needs to be determined in large prospective biomarkers studies, though first results are promising.

Major:

Additional information on the immunoassay should be provided:

The authors defined a cut-off value of > 0.2 % CD45+ cells for tumor sample analyses. How was this threshold defined? It would be very helpful to visualize CD45+ infiltration by immunohistochemistry in some analyzed tissues, showing examples with high and low infiltrates that fulfill the assay criteria.

Tumor mutational burden is a response biomarker in anti-PD1 therapy of NSCLC. Do NSCLC samples of the Rclus group show a higher TMB compared to the NRclus group?

Minor:

The authors should explain why defucosylated anti-CTLA-4 antibody has been used

Fig. 4A, some "n" (vertically) are not correctly counted.

Referee #3 (Remarks for Author):

The manuscript EMM-2020-12850 entitled 'Immunodynamics of Explanted Human Tumors for Precision and Personalized Immuno-Oncology' reports the development and application of an in vitro platform enabling dynamic multiplexed immunophenotyping.

Cancer specimens (> 2cm) from various tumor types were dissociated and used for phenotyping and flow cytometry analysis at day 0 and/or incubated with immune checkpoint inhibitors and immunostimulatory reagents. The readout was performed after three days by flow cytometry and a beads-based multiplex immunoassay. Tumor supernatants from resected cancer specimens were used to assess soluble factors in the same manner.

The in vitro assay allowed to segregate 43 tested tumor samples into two clusters of responders and non-responders towards anti-PD1. Differences between R and NR were described in detail. Moreover, the IRS was introduced to further classify samples based on TCR-dependent soluble factors. Results of the IRS classification system mostly aligned with the non-supervised hierarchical clustering method. The in vitro assay predicted 5/7 anti-PD1 responses correctly.

Next, soluble factors from the supernatant of resected tumors were associated with a) TIL phenotype after anti-PD1 treatment b) IRS and c) phenotypes at day 0. By doing so CXCL10 was identified and confirmed as predictive biomarker of anti-PD1 response.

The in vitro assay was also used to test anti-PD1 in combination with other I-O compounds. 26.3% of anergic samples and 50% of IRSlow samples were shifted towards a more immunogenic profile after combinatorial treatment.

Lastly the HRS was used to identify hyporesponsive tumors towards PD1 blockade.

Overall the manuscript describes a novel and versatile assay that allows rapid assessment of immunophenotypes in various cancer types as well as the immune response towards immune checkpoint inhibitors. Given the relatively high percentage of patients not responding to immunotherapy, it is of high significance to develop personalized systems that allow assessment of the tumor immune response towards current agents, like anti-PD1, and novel agents. The assay has the potential to possibly predict immune responses and evaluate which combinatorial treatments could increase immunogenicity. It was stated that many findings confirmed current literature, hence, showing that the assay recognizes the already reported findings. All findings are well represented and coherent with the data that is included. The manuscript certainly contains novel claims and frames them into the context of earlier literature. Nonetheless, there is a risk of overselling the assay and findings should be further evaluated by for example more mechanistic experiments when looking at the combinatorial treatment claims, increased sample size or more patient data in respect to the predictive value of the assay. Patient numbers and diversity of tumors and outcomes prohibit strong conclusions and the authors should more describe this is a first step into a new methodology. A limitation for future applications is the sizeable tumor fragment needed, high exclusion rate of tumor samples at the start and therefore a potential bias in the readout. The established assay is superior to other models in terms of versatility, dynamic and timing. It allows a rapid readout of a diverse set of immune-relevant parameters and has the potential to address many relevant questions that could help improve making treatment decisions.

Major concerns

1.

a. One major concern is that only around 36% of tumor samples qualify for further processing (Supplementary Table S2). Even though I agree with the selection criteria this makes it hardly possible to use the assay in a clinical setting as a predictive tool or to use it for immediate treatment decisions. It would be great if the author could comment on this and discuss how they will improve the assay to achieve a higher rate of tumor samples and also of smaller sizes (biopsies).

b. In addition to 1.1. it is important to realize that the eligibility criteria for tumor sample selection create a biased selection of samples since for example "immune scarce" tumor will be excluded. This is unlikely to interfere with the actual result of a single tumor sample but concluding from the results of the selected tumors to a broader spectrum has to be done carefully.

2. Resistance towards anti-PD1 was predicted in 5/7 cases (Supplementary Table S4). Since 6/7 are NR, as defined by the in vitro assay, it is difficult to evaluate the predictive value of the assay. It would be optimal if in vitro assay R would also be included and assessed. By showing that the assay aligns both ways, R and NR, with clinical outcome the claim of a predictive assay would be more solid. As presented here, with exception of the special case K11, the assay seems to correctly predict the NR cases but it is unclear if the assay holds up for R as well and in a larger sample size.

As stated by the authors, the sample size is small and would need to be expanded to address this and provide more substance to their claims.

Minor concerns

3. One highlight of the manuscript is the potential use of the assay for evaluation and testing of combinatorial treatments (Figure 3). Promising results are presented but it is key to confirm those findings by further experiments. Tumor recognition and killing by T cells could be evaluated under influence of certain combinations to proof higher immunogenicity.

4. The evaluation of hypo-responsiveness towards anti-PD1 is well presented (Expanded View Figure 4) and valuable to present the versatility of the assay. But to link it to hyper-progression further experiments are necessary.

5. The concept of analyzing the supernatant of resection samples is simplistic and straight forward. The results presented are, to some extent, inconclusive which could indicate that the approach is too rudimental. It is possible that soluble factors are also tissue specific and maybe healthy resection supernatant, in theory, would be necessary to evaluate a baseline.

In summary the manuscript 'Immunodynamics of Explanted Human Tumors for Precision and Personalized Immuno-Oncology' presents some interesting early observations but it requires larger numbers of patients to validate their findings. The manuscript could be better positioned as a brief methodology paper. The established in vitro assay has potential to address relevant immunotherapeutic questions and could potentially lead to medical relevant improvements. But it has to be evaluated to what extend the assay can be used in a clinical set up and also the predictive value needs to be further elaborated.

EMM-2020-12850 Point-by-point reply Dubuisson et al.

POINT-BY-POINT REPLY

Editor comments

"A major criticism shared by the referees is the small number of samples, which limits the predictive value of the assay. Therefore, more samples should be included in the study. Alternatively, the referees agree that the assay by itself is of interest, and would agree for the data to be published as a shorter report should you not be able to provide more samples. In that case, the claims should be toned down accordingly".

******* Reviewer's comments *******

Referee #1

Well designed and conducted study, using human samples. Dubuisson et al. established an "*in vitro*" (ex vivo/in situ) assay to analyze the immune reactivity of cancer infiltrates within primary tumors to various immune checkpoint inhibitors and activators. The approach looks very interesting.

We thank the referee for this appreciation.

However, the authors need to explain in more detail the experimental set-up (dynamic multiplexed immunophenotyping assay).

Our response:

In the revised manuscript written as a "Brief Report", we provided a more detailed graphical abstract in Figure 1A. and elaborated a comprehensive description of the dynamic multiplexed immunophenotyping assay in material and methods section.

1. It is recommended to do a parameter study to show the calibration and robustness of the method. For example, they stated "A tumor was considered immune reactive (or IRS^{high}) when the immune reactivity score IRS ≥ 41.2 ". is there any reference for this exact value?

Our response:

Sorry for the lack of clarity. In Figure EV1. panel C., we already provided the mathematical rationale for the calibration of the assay and the choice of this cut-off value as the one maximizing the positive likelihood ratio of response to PD-1 blockade.

-In order to evaluate the stability of this cut-off value, we ran 2000 bootstraps and now report the frequency of each cut-off maximizing the positive likelihood ratio (Fig.1A of this point by point reply).

-The non-supervised hierarchical clustering analysis presented in Fig.1 B displays two clusters of tumor immune response after anti-PD-1 *in vitro* stimulation, with opposite capacities to be immunomodulated by anti-PD-1 mAbs (classified as "R_{clust}" or NR_{clust}). These clusters have Approximately Unbiased (AU) p-values above 95% after 10 000 multi-scale bootstrap resampling (Fig.2 of this point by point reply).

-Then, as we wanted to provide a quantitative surrogate for this hierarchical clustering, we developed the Immune Response Score (IRS) that represents the percentage of T cell dependent- cytokines increased by at least a 1.5 log₂ fold ratio post-stimulation. The fitness of

this score to the Rclustering is strong (AUC = 0.9792 [0.9325-1]). As outlined previously, the cut-off of 41.18 has been shown to maximize the positive likelihood ratio.

-Finally, to further confirm the robustness of this IRS score, we worked in collaboration with the Department of onco-dermatology of Gustave Roussy conducting the NIVIPIT clinical trial (<https://clinicaltrials.gov/show/NCT02857569>) aimed at randomizing intratumor (IT) versus systemic (IV) administration of anti-CTLA-4 mAbs in conjunction with systemic anti-PD-1 mAbs in first line metastatic melanoma patients. *In situ* assay was applied to the initial macro-biopsy of one metastatic (often subcutaneous) lesion, evaluating 9/17 available TCR-dependent cytokines. Of note, those 9 cytokines allowed us to find the same IRS than the one calculated with the 17 TCR-dependent cytokines as shown in figure S1. Using the same IRS cut-off value we could prospectively predict the response or resistance to anti-PD-1+anti-CTLA-4 co-blockade in 6/8 cases (3 responders, 1 stable disease, 2 non responders) as shown in Figure 3 of this point-by-point reply. This assay is currently being implemented in all of the 45 patients enrolled in this trial, and results are considered preliminary. However, these results lend support to the robustness of the assay. These data will not be included in the revised version of the manuscript that will be recrafted as a "brief report", highlighting the fact that the results need to be prospectively validated using downscaling process and tumor biopsies (rather than whole tumor pieces).

Figure 2: Tumor classification according to IRS.

Tumors are hierarchically clustered according to the normalized log2 fold ratio of 27 soluble factors secreted after PD-1 blockade as in Fig.1b. Approximately Unbiased (AU) p-values for each cluster are computed by 10 000 multiscale bootstrap resampling. Clusters with AU larger than 95% are highlighted by red rectangles, which are strongly supported by data.

Figure for referees removed

Figure S1. IRS downscaling from 17- to 9- TCR dependent cytokines.

From our cohort of 42 resectable tumors, 9-TCR dependent cytokines used to evaluate biopsie's IRS and responses to treatments (9 cytokines IRS, y-axis) are shown here to correspond to the IRS performed with 17-TCR dependent cytokines (Complete IRS, x-axis)

2. Tumor supernatants were produced by both tumor cells and TILs. How did the authors unsure that the tumor sample were all of the same size or same weight? Please provide the size or weight of each samples.

Our response:

This point is very well taken. We have indeed recorded the weight of the whole specimen for 22 tumors as well as the protein content of the tumor supernatant using the Bradford method. We normalized each cytokine/chemokine concentration to that of the overall protein content of each specimen. We provide -in the revised brief report in Appendix Table S2- the details of weight/protein content of the Bradford assay (refer to Table 1 of this PBPR).

Supporting this contention, we did not find significant differences between the tumor weight and our best clinical correlate *i.e* CXCL10 release. Moreover, IRS was evenly represented across tumor weight as shown in PBPR Figure 4.

Type Tumoral	New ID	Tumor Weight (g)	Total protein concentration in tumor supernatant (µg/mL)
Lung	L5	2,10	2629,18
Lung	L7	4,20	4205,79
Lung	L8	3,42	6526,31
Lung	L9	1,90	6748,37
Lung	L10	2,41	1765,37
Lung	L12	2,01	2553,68
Lung	L13	1,78	1943,01
Lung	L14	1,02	2464,85
Kidney	K4	6,57	136,15
Kidney	K5	0,74	575,13
Kidney	K7	1,62	2829,03
Kidney	K8	2,50	3872,71
Kidney	K9	5,11	4094,76
Kidney	K10	3,60	985,94
Kidney	K11	4,40	1296,82
Kidney	K15	16,27	13476,75
Kidney	K16	1,91	5926,75
Kidney	K17	6,87	6670,65
Kidney	K18	3,81	7425,65
Kidney	K19	8,55	3106,60
Head & Neck	HN3	0,52	1097,00
Head & Neck	HN5	1,01	1907,49

Table 1 (included in the Table S2 of the revised manuscript). List of each sample indicating weight (when available), and protein content by Bradford assay.

Figure 4. Absence of impact of tumor weight on both CXCL10 release and IRS. Bar graph separating all tumors into 3 groups of increasing weight. Each dot represents one tumor and the binar color code of the dots refer to IRS high (red) or low (purple).

3. Why did the incubation period range from 2 to 24 hours? Does incubation period affect the cytokines concentration?

Our response:

The time elapsing from the withdrawal of the tumor sample from the operating room to the pathologist and then to the lab varied from 2 to 18 hours depending on surgical constraints

(Figure 5 of the PBPR). We did our best to limit this time lapse. However, only one tumor supernatant was harvested almost immediately while the vast majority was within 6-18hrs.

Figure 5. CXCL10 release in two groups according to time elapsing between the operating room and tumor processing.

Each dot represents one tumor and the binary color code of the dots refer to IRS high (red) or low (purple).

4. The clinical data (S-table4) is very limited. This should be acknowledged and discussed as a limitation.

Our response:

Indeed, we will acknowledge this limitation due to the retrospective nature of the analysis over 3 years. A prospective validation is mandatory, that will be facilitated by the manuscript acceptance for publication.

Referee #2 (Remarks for Author):

With the new immunoassay and immune reactivity score the authors developed a broadly applicable platform to analyze tumor responses to single and combined anti-PD1-based treatments in vitro and guide therapy decisions in immuno-oncology. Data from a limited sample set (n=7) suggest that this assay might be suitable also in predicting response to anti-PD1 therapy. Overall, this assay format will strongly improve translational studies in immuno-oncology. Its predictive value needs to be determined in large prospective biomarkers studies, though first results are promising.

We thank the referee for his/her interest in this assay.

Major: Additional information on the immunoassay should be provided: The authors defined a cut-off value of > 0.2 % CD45+ cells for tumor sample analyses. How was this threshold defined? It would be very

helpful to visualize CD45+ infiltration by immunohistochemistry in some analyzed tissues, showing examples with high and low infiltrates that fulfill the assay criteria.

Our response:

Indeed, we made sure that this flow cytometry-based threshold estimate corresponded to about 10-30% TIL infiltration (meaning about 100-200/mm² in the stroma) by pathologist-based CD45+immunohistochemistry for the vast majority of the specimen, in accordance with prior work reported in clinically relevant immunoscores of breast (Denkert C et al. PMID: 25534375 Clinical Trial. Loi S et al. *J Clin Oncol.* 2019 Mar 1;37(7):559-569) and colon cancers (Pagès F et al. PMID: 30496076, Pagès F et al. PMID: **16371631**, Pagès F et al PMID: 19858404). We have evaluated 12 available tumors/biopsies in our cohort. Figure 6 of this point-by-point reply summarizes the positive correlation between flow cytometry-based TILs percentages and IHC-based TILs infiltrates, justifying our cut-off of 0.2%. We have included in the revised Fig. EV1 panel A, two representative panels of CD45 staining in high versus low TIL infiltrates, within the tumor nests and the stroma (2 micrograph pictures at high magnification power field).

Figure 6. Positive correlation confirming the flow cytometry-based cut-off value of TIL 0.2% using CD45 immunostainings of tumor biopsies. Spearman correlation between flow-cytometry-based TILs infiltrates and IHC-based TILs infiltrate in stroma.

Tumor mutational burden is a response biomarker in anti-PD1 therapy of NSCLC. Do NSCLC samples of the Rclus group show a higher TMB compared to the NRclus group?

Given that TMB did not get the formal EMA approval for immune checkpoint inhibitors (ICB) outside Big pharma-driven clinical trials, we have not been able to systematically perform the TMB assessment in most cases where ICB were not even administered.

Minor:

The authors should explain why defucosylated anti-CTLA-4 antibody has been used Fig. 4A.

Our response:

Indeed, FcγR-mediated antibody-dependent cytotoxicity facilitates the elimination of the cells expressing the Fab'2 binding antigen of the antibody, enabling the killing of CTLA4^{high} regulatory T cells by intratumoral CD16 (FcγR)+ NK or macrophages (apostrophed the

"ADCC" effect). Defucosylation of the Fc portion of an antibody theoretically increases the ADCC effect (reviewed in Picardo S et al. PMID: **31877721**). The clinical relevance of the ADCC mechanism *i.e* the FCGR3A and FCGR2A single nucleotide polymorphisms, has been brought up in clinical trial utilizing ipilimumab/ tremelimumab anti-CTLA4 mAb (Arce Vargas F et al. PMID: 29576375), rituximab, or trastuzumab for instance (Cartron G, *Blood* 2002;99:754–8, Gavin PG et al. PMID: 27812689). We have briefly explained the concept in page 11 of the revised manuscript.

Some "n" (vertically) are not correctly counted.

Sorry. We have modified these errors.

Referee #3 (Remarks for Author):

Overall the manuscript describes a novel and versatile assay that allows rapid assessment of immunophenotypes in various cancer types as well as the immune response towards immune checkpoint inhibitors. ...The assay has the potential to possibly predict immune responses and evaluate which combinatorial treatments could increase immunogenicity. It was stated that many findings confirmed current literature, hence, showing that the assay recognizes the already reported findings. All findings are well represented and coherent with the data that is included. The manuscript certainly contains novel claims and frames them into the context of earlier literature.... The established assay is superior to other models in terms of versatility, dynamic and timing. It allows a rapid readout of a diverse set of immune-relevant parameters and has the potential to address many relevant questions that could help improve making treatment decisions.

We thank the referee for his/her interest in this assay.

Nonetheless, there is a risk of overselling the assay and findings should be further evaluated by for example more mechanistic experiments when looking at the combinatorial treatment claims, increased sample size or more patient data in respect to the predictive value of the assay. Patient numbers and diversity of tumors and outcomes prohibit strong conclusions and the authors should more describe this is a first step into a new methodology. A limitation for future applications is the sizeable tumor fragment needed, high exclusion rate of tumor samples at the start and therefore a potential bias in the readout.

Our response:

We agree with these drawbacks. Therefore, we decided to assess the prediction of response to PD-1 blockade and train the model system in a small independent study scaling down the analysis on biopsies instead of large/whole tumor piece. The results are shown in Figure 3 of this point-by-point reply and discussed below. However, given that we could not -stricto sensu- add more specimen handled under the same procedure (using various combinatorial

regimen in parallel) during the COVID-19 health crisis to broaden the exemplifications, we decided to reformat the manuscript in a smaller article, such as a "Brief report" and to tone down the predictive power of the method, as advised by the 3 referees and the conclusive remark of the editorial board.

Major concerns

1a. One major concern is that only around 36% of tumor samples qualify for further processing (Supplementary Table S2). Even though I agree with the selection criteria this makes it hardly possible to use the assay in a clinical setting as a predictive tool or to use it for immediate treatment decisions. It would be great if the author could comment on this and discuss how they will improve the assay to achieve a higher rate of tumor samples and also of smaller sizes (biopsies).

Our response:

These two points (TIL content, and downscaling to biopsies for the assay) are very well taken. First of all, the cut-off value of >0.2% CD45+ (that mostly explain the feasibility rate of 36%) is based on several factors (surgical piece too small for that many conditions, restrictions or constraints from the pathologists, ethical guidelines for biopsy sizes...)

Indeed, we made sure that this flow cytometry-based threshold estimate corresponded to about 10-30% TIL infiltration (meaning about 10-20 CD45+ cells/mm² within the tumor nests and 100-200/mm² in the stroma) by pathologist-based CD45+immunohistochemistry for the vast majority of the specimen (12 tumors/biopsies could be evaluated), in accordance with prior work reported in clinically relevant immunoscores of breast (Denkert C et al. PMID: 25534375 Clinical Trial. Loi S et al. **J Clin Oncol.** 2019 Mar 1;37(7):559-569) and colon cancers (Pagès F et al. PMID: 30496076, Pagès F et al. PMID: **16371631**, Pagès F et al PMID: 19858404). Figure 6 of the point-by-point reply (refer above, Referee 1) of this point-by-point reply summarizes correlation between flow cytometry-based TILs percentages and IHC-based TILs infiltrates. We have included in the revised Fig. EV1 panel A, two representative panels of CD45 staining in high versus low TIL infiltrates, within the tumor nests and the stroma (4 micrograph pictures at high magnification power field).

Hence, we could prospectively improve these sampling limitations in future trials to validate the clinical relevance of these findings by:

- obtaining better macro-biopsies or several biopsies from different accessible sites for each patient (specifically in melanoma or lymphoma etc...)
- limiting the number of experimental conditions (for instance all the isotype control antibodies, immune checkpoint of limited significance (IDO inhibitors, GITR blockade etc...)
- downsizing the number of cells per well (since chemokine/cytokine monitoring require less cells than flow-cytometry based assays)

Despite these limitations, and to further support the robustness of the IRS score using limited amount of tissues such as regular "biopsies", we worked in collaboration with the Department of onco-dermatology of Gustave Roussy conducting the NIVIPIT clinical trial (<https://clinicaltrials.gov/show/NCT02857569>) aimed at randomizing intratumor (IT) versus systemic (IV) administration of anti-CTLA-4 mAb in conjunction with systemic anti-PD-1 mAbs in first line metastatic melanoma patients. *In situ* assay was

applied to the initial macro- biopsy of one metastatic (often subcutaneous) lesion, evaluating 9/17 available TCR-dependent cytokines. Of note, those 9 cytokines allowed us to find the same IRS than the one calculated with the 17 TCR-dependent cytokines as shown in figure S1. Using the same IRS cut-off value we could prospectively predict the response or resistance to anti-PD-1+anti-CTLA-4 co-blockade in 6/8 cases (3 responders, 1 stable disease, 2 non responders) as shown in Figure 3 of this point-by-point reply (refer to referee 1, above). This assay is currently being implemented in all of the 45 patients, and results are considered preliminary but support the robustness of this assay. Therefore, these data will not be included in the revised version of the manuscript that will be recrafted as a "brief report", highlighting the fact that the results need to be prospectively validated using downscaling process and tumor biopsies (rather than whole tumor pieces).

1b. In addition to 1.1. it is important to realize that the eligibility criteria for tumor sample selection create a biased selection of samples since for example "immune scarce" tumor will be excluded. This is unlikely to interfere with the actual result of a single tumor sample but concluding from the results of the selected tumors to a broader spectrum has to be done carefully.

We agree and toned down our conclusions.

2. Resistance towards anti-PD1 was predicted in 5/7 cases (Supplementary Table S4). Since 6/7 are NR, as defined by the in vitro assay, it is difficult to evaluate the predictive value of the assay. It would be optimal if in vitro assay R would also be included and assessed. By showing that the assay aligns both ways, R and NR, with clinical outcome the claim of a predictive assay would be more solid. As presented here, with exception of the special case K11, the assay seems to correctly predict the NR cases but it is unclear if the assay holds up for R as well and in a larger sample size. As stated by the authors, the sample size is small and would need to be expanded to address this and provide more substance to their claims.

Our response:

As shown above on 8 additional prospective melanoma patients (Figure 3), the *in vitro* assay could be accurately predictive of responders in 3/5 cases. Prospective enrolment will allow a firm conclusion in a near future.

Minor concerns

3. One highlight of the manuscript is the potential use of the assay for evaluation and testing of combinatorial treatments (Figure 3). Promising results are presented but it is key to confirm those findings by further experiments. Tumor recognition and killing by T cells could be evaluated

under influence of certain combinations to proof higher immunogenicity.

Our response:

Indeed, it would be possible to evaluate apoptosis of tumor cells (cleaved caspase3 stainings or annexin V staining on tumor cells) or CTL degranulation (membrane CD107A) at early time points of 18 hours post-stimulation.

4.The evaluation of hypo-responsiveness towards anti-PD1 is well presented (Expanded View Figure 4) and valuable to present the versatility of the assay. But to link it to hyper-progression further experiments are necessary.

Our response:

This point is very well taken. We followed the editor's recommendation to shorten the manuscript into a "brief report", to limit the emphasis on this part by placing it in the Figure EV4.

5.The concept of analyzing the supernatant of resection samples is simplistic and straight forward. The results presented are, to some extent, inconclusive which could indicate that the approach is too rudimental. It is possible that soluble factors are also tissue specific and maybe healthy resection supernatant, in theory, would be necessary to evaluate a baseline.

Our response:

We agree with these comments. We analyzed CXCL10 release in 5 non tumor lung fragments. Figure 7 highlights the baseline level being not greater than 10^{-7} and 10 times less than the mean of IRS low or 100 times less than the mean of IRS high.

Each dot represents one sample and the color code of the dots refer to IRS high (red) or IRS low (purple) tumor samples and non tumor (light blue) specimens.

In summary the manuscript 'Immunodynamics of Explanted Human Tumors for Precision and Personalized Immuno-Oncology' presents some interesting early observations but it requires larger numbers of patients to validate their findings. The manuscript could be better positioned as a brief methodology paper. The established in vitro assay has potential to address relevant immunotherapeutic questions and could potentially lead to medical relevant improvements. But it has to be evaluated to what extend the assay can be used in a clinical set up and also the predictive value needs to be further elaborated.

As outlined above, we followed the referee and the editor's recommendation to shorten the manuscript into a "brief report" and to tone down the conclusions.

19th Oct 2020

Dear Prof. Zitvogel,

Thank you for the submission of your revised manuscript to EMBO Molecular Medicine. We have now received the enclosed reports from the two referees who reviewed the new version of your manuscript. As you will see, they are supportive of publication, and I am thus pleased to inform you that we will be able to accept your manuscript pending the following final minor amendments:

1) Main manuscript text:

- Please answer/correct the changes suggested by our data editors in the main manuscript file (in track changes mode). Please use this file for any further modification.
- We can accommodate up to 5 keywords, please update accordingly.
- Please remove the green highlighted text.
- Please update the reference format so as to have 10 authors before et al. In the main text, references should appear as (Author et al, year).
- The "Summary" should be renamed "The paper explained".
- Material and methods:
 - o Human cohorts: Please include a statement that the experiments conformed to the principles set out in the WMA Declaration of Helsinki and the Department of Health and Human Services Belmont Report. (This statement should also be included in the checklist).
 - o Immunohistochemistry: "according to standard protocols": please give enough information so that the reader can reproduce the experiment.
 - o Please provide the antibody dilutions used for staining.
 - Indicate in legends exact n= and p= values, not a range, along with the statistical test used (including for p values that are not significant, ns). Some people found that to keep the figures clear, providing a supplemental table with all exact p-values was preferable. You are welcome to do this if you want to.
 - Author contribution: Nicolas Jacquelot is missing, please update.
 - Please note that we now mandate that all corresponding authors list an ORCID digital identifier (missing for Anne-Gaëlle Goubet). We note that you currently have together with you, a total of 4 co-corresponding authors. Do you confirm equal contribution of these 4 people, able to take full responsibility for the paper and its content? While there is no limit per se to the number of co-corresponding authors, 3 is rare, 4 even more so, and may not reflect as intended to the community.
 - Please update the EV callouts from Expanded View Figure 1 to Figure EV1 etc
 - Please include a Data availability section:

Before submitting your revision, primary datasets produced in this study need to be deposited in an appropriate public database (see <https://www.embopress.org/page/journal/17574684/authorguide#dataavailability>). If not applicable, the following sentence should be included: "This study includes no data deposited in external repositories".

Information in Data availability should be reflected in the checklist (if no datasets was introduced, please indicate N/A in section F19).

2) Source Data:

Thank you for providing Source Data. Please upload them so as to have one file/figure for the main figures, and one file for all EV figures. Please annotate the pictures provided for Fig. EV1.

3) Appendix:

The resolution of several appendix figures is low; please provide higher resolution figures where possible.

4) For more information: There is space at the end of each article to list relevant web links for further consultation by our readers. Could you identify some relevant ones and provide such information as well? Some examples are patient associations, relevant databases, OMIM/proteins/genes links, author's websites, etc...

5) Thank you for providing a synopsis. I slightly modified the text to fit our style and format, please let me know if you agree with the following:

To predict cancer resistance to PD-1 blockade and design suitable combinations of immunomodulators, a 60-hours functional in vitro assay was set up in 43 tumors that allowed calculation of the "Immune Reactivity Score (IRS)" based on 17 TCR-dependent-cytokines/chemokines.

- Primary resistance to PD-1 blockade could be predicted in vitro and in vivo with ~ 70% accuracy
- CXCL10 was the best in situ predictor of IRS score and response to PD-1 blockade in patients
- 50% of primary resistance to PD-1 blockade could be overcome by a personalized combinatorial regimen
- Hypo-responders to PD-1 blockade could be prevented by combining anti-PD-1 and anti-KIR mAbs.

Thank you for providing a nice synopsis image. Please make sure that the text remains readable when resized to 550px wide.

6) As part of the EMBO Publications transparent editorial process initiative (see our Editorial at <http://embomolmed.embopress.org/content/2/9/329>), EMBO Molecular Medicine will publish online a Review Process File (RPF) to accompany accepted manuscripts.

In the event of acceptance, this file will be published in conjunction with your paper and will include the anonymous referee reports, your point-by-point response and all pertinent correspondence relating to the manuscript. We note that you want to exclude Figure 3, is that correct?

I look forward to receiving your revised manuscript.

Yours sincerely,

Lise Roth

Lise Roth, PhD
Editor
EMBO Molecular Medicine

To submit your manuscript, please follow this link:

Link Not Available

The system will prompt you to fill in your funding and payment information. This will allow Wiley to send you a quote for the article processing charge (APC) in case of acceptance. This quote takes into account any reduction or fee waivers that you may be eligible for. Authors do not need to pay any fees before their manuscript is accepted and transferred to our publisher.

***** Reviewer's comments *****

Referee #2 (Remarks for Author):

The authors adequately addressed the points I raised.

Referee #3 (Remarks for Author):

Dubuisson et al. adapted their manuscript EMM-2020-12850-V2 according to all remarks and responded to the referee's criticism in an adequate way. We agree with their decision to shorten the manuscript and publish the results as a short report since sample size could not be increased. The presented in vitro assay has the potential to be of great interest in the field of personalized immune-oncology as a prediction tool but, as the author agrees, needs to be validated. Therefore we appreciate the authors effort to tone down their conclusions and support the current version of the manuscript.

EMM-2020-12850 Final amendments responses Dubuisson et al.

1) Main manuscript text:

- Please answer/correct the changes suggested by our data editors in the main manuscript file (in track changes mode). Please use this file for any further modification.

Our response:

Changes have been made according to editor's comments in the main manuscript file

- We can accommodate up to 5 keywords, please update accordingly.

Our response:

We have updated the 5 keywords accordingly: cancer, precision oncology, immune checkpoint inhibitors, "*in vitro*" assay, immunomonitoring.

- Please remove the green highlighted text.

Our response:

Green highlighted texts have been removed from the main manuscript file.

- Please update the reference format so as to have 10 authors before et al. In the main text, references should appear as (Author et al, year).

Our response:

Sorry for this mistake, references have now been changed accordingly.

- The "Summary" should be renamed "The paper explained".

Our response:

We have renamed it.

- Material and methods:

o Human cohorts: Please include a statement that the experiments conformed to the principles set out in the WMA Declaration of Helsinki and the Department of Health and Human Services Belmont Report. (This statement should also be included in the checklist).

Our response:

This statement has been included in both the main manuscript file and in the author checklist.

o Immunohistochemistry: "according to standard protocols": please give enough information so that the reader can reproduce the experiment.

Our response:

Additional information has been included in the main manuscript file.

o Please provide the antibody dilutions used for staining.

Our response:

Antibody dilutions for IHC have been included in the main manuscript file. Flow cytometry antibody dilution is indicated at the end of section C6 in the author checklist.

- Indicate in legends exact n= and p= values, not a range, along with the statistical test used (including for p values that are not significant, ns). Some people found that to keep the figures clear, providing a supplemental table with all exact p-values was preferable. You are welcome to do this if you want to.

Our response:

We provided now a supplemental table with all exact number of specimen (n) and p-values.

- Author contribution: Nicolas Jacquelot is missing, please update.

Our response:

Sorry for this mistake, we have now added N Jacquelot's contribution to this work.

- Please note that we now mandate that all corresponding authors list an ORCID digital identifier (missing for Anne-Gaëlle Goubet). We note that you currently have together with you, a total of 4 co-corresponding authors. Do you confirm equal contribution of these 4 people, able to take full responsibility for the paper and its content? While there is no limit per se to the number of co-corresponding authors, 3 is rare, 4 even more so, and may not reflect as intended to the community.

Our response:

Prof. Laurence Zitvogel is the only corresponding author of this paper. Agathe Dubuisson, Jean-Eudes Fahrner and Anne-Gaëlle are co-first authors but not co-corresponding authors.

- Please update the EV callouts from Expanded View Figure 1 to Figure EV1 etc

Our response:

Sorry for this mistake, EV callouts have now been changed accordingly.

- Please include a Data availability section:

Before submitting your revision, primary datasets produced in this study need to be deposited in an appropriate public database (see

<https://www.embopress.org/page/journal/17574684/authorguide#dataavailability>). If not applicable, the following sentence should be included: "This study includes no data deposited in external repositories".

Information in Data availability should be reflected in the checklist (if no datasets was introduced, please indicate N/A in section F19).

Our response:

We have included a new data availability section in the main manuscript file where we stated "This study includes no data deposited in external repositories" and reflected it with an N/A in section F19 of the author checklist.

2) Source Data:

Thank you for providing Source Data. Please upload them so as to have one file/figure for the main figures, and one file for all EV figures. Please annotate the pictures provided for Fig. EV1.

Our response:

We have now provided Source Data accordingly: one excel file for main figure 2 and one pdf file for all EV figures.

3) Appendix:

The resolution of several appendix figures is low; please provide higher resolution figures where possible.

Our response:

Sorry for this mistake, we have improved the resolution quality figures of all appendix figures.

4) For more information: There is space at the end of each article to list relevant web links for further consultation by our readers. Could you identify some relevant ones and provide such information as well? Some examples are patient associations, relevant databases, OMIM/proteins/genes links, author's websites, etc...

Our response:

We have included a relevant website section in the main manuscript file where we included Prof. Laurence Zitvogel lab's website.

5) Thank you for providing a synopsis. I slightly modified the text to fit our style and format, please let me know if you agree with the following:

To predict cancer resistance to PD-1 blockade and design suitable combinations of

immunomodulators, a 60-hours functional in vitro assay was set up in 43 tumors that allowed calculation of the "Immune Reactivity Score (IRS)" based on 17 TCR-dependent-cytokines/chemokines.

- Primary resistance to PD-1 blockade could be predicted in vitro and in vivo with ~ 70% accuracy
- CXCL10 was the best in situ predictor of IRS score and response to PD-1 blockade in patients
- 50% of primary resistance to PD-1 blockade could be overcome by a personalized combinatorial regimen
- Hypo-responders to PD-1 blockade could be prevented by combining anti-PD-1 and anti-KIR mAbs.

Our response:

Thank you for these changes, we have included the modification in the Paper's synopsis word file.

Thank you for providing a nice synopsis image. Please make sure that the text remains readable when resized to 550px wide.

Our response:

We checked that the text remains readable when synopsis image is resized to 550px wide.

6) As part of the EMBO Publications transparent editorial process initiative (see our Editorial at <http://embomolmed.embopress.org/content/2/9/329>), EMBO Molecular Medicine will publish online a Review Process File (RPF) to accompany accepted manuscripts.

In the event of acceptance, this file will be published in conjunction with your paper and will include the anonymous referee reports, your point-by-point response and all pertinent correspondence relating to the manuscript. We note that you want to exclude Figure 3, is that correct?

Our response:

Thank you for this consideration, we confirm that we want to exclude Figure 3 of the point-by-point responses of the referee.

28th Oct 2020

Dear Prof. Zitvogel,

We are pleased to inform you that your manuscript is now accepted for publication and will be sent to our publisher to be included in the next available issue of EMBO Molecular Medicine.

Please read below for additional IMPORTANT information regarding your article, its publication and the production process.

Congratulations on your interesting work!

With my best wishes,

Lise Roth

Lise Roth, Ph.D
Scientific Editor
EMBO Molecular Medicine

Follow us on Twitter @EmboMolMed
Sign up for eTOCs at embopress.org/alertsfeeds

*** ** IMPORTANT INFORMATION ** **

SPEED OF PUBLICATION

The journal aims for rapid publication of papers, using the advance online publication "Early View" to expedite the process: A properly copy-edited and formatted version will be published as "Early View" after the proofs have been corrected. Please help the Editors and publisher avoid delays by providing e-mail address(es), telephone and fax numbers at which author(s) can be contacted.

Should you be planning a Press Release on your article, please get in contact with embomolmed@wiley.com as early as possible, in order to coordinate publication and release dates.

LICENSE AND PAYMENT:

All articles published in EMBO Molecular Medicine are fully open access: immediately and freely available to read, download and share.

EMBO Molecular Medicine charges an article processing charge (APC) to cover the publication costs. You, as the corresponding author for this manuscript, should have already received a quote

with the article processing fee separately. Please let us know in case this quote has not been received.

Once your article is at Wiley for editorial production you will receive an email from Wiley's Author Services system, which will ask you to log in and will present you with the publication license form for completion. Within the same system the publication fee can be paid by credit card, an invoice, pro forma invoice or purchase order can be requested.

Payment of the publication charge and the signed Open Access Agreement form must be received before the article can be published online.

PROOFS

You will receive the proofs by e-mail approximately 2 weeks after all relevant files have been sent to our Production Office. Please return them within 48 hours and if there should be any problems, please contact the production office at embopressproduction@wiley.com.

Please inform us if there is likely to be any difficulty in reaching you at the above address at that time. Failure to meet our deadlines may result in a delay of publication.

All further communications concerning your paper proofs should quote reference number EMM-2020-12850-V3 and be directed to the production office at embopressproduction@wiley.com.

Thank you,

Lise Roth, Ph.D
Scientific Editor
EMBO Molecular Medicine

Corresponding Author Name: Prof. Laurence Zitvogel

Manuscript Number: EMM-2020-12850